# NOT ALL LANGUAGE MODEL FEATURES ARE ONE-DIMENSIONALLY LINEAR

**Joshua Engels**
MIT
jengels@mit.edu

**Eric J. Michaud**
MIT & IAIFI
ericjm@mit.edu

**Isaac Liao**
MIT
iliao@mit.edu

**Wes Gurnee**
MIT
wesg@mit.edu

**Max Tegmark**
MIT & IAIFI
tegmark@mit.edu

## ABSTRACT

Recent work has proposed that language models perform computation by manipulating one-dimensional representations of concepts ("features") in activation space. In contrast, we explore whether some language model representations may be inherently multi-dimensional. We begin by developing a rigorous definition of irreducible multi-dimensional features based on whether they can be decomposed into either independent or non-co-occurring lower-dimensional features. Motivated by these definitions, we design a scalable method that uses sparse autoencoders to automatically find multi-dimensional features in GPT-2 and Mistral 7B. These auto-discovered features include strikingly interpretable examples, e.g. *circular* features representing days of the week and months of the year. We identify tasks where these exact circles are used to solve computational problems involving modular arithmetic in days of the week and months of the year. Next, we provide evidence that these circular features are indeed the fundamental unit of computation in these tasks with intervention experiments on Mistral 7B and Llama 3 8B, and we examine the continuity of the days of the week feature in Mistral 7B. Overall, our work argues that understanding multi-dimensional features is necessary to mechanistically decompose some model behaviors.

## 1 INTRODUCTION

Language models trained for next-token prediction on large text corpora have demonstrated remarkable capabilities, including coding, reasoning, and in-context learning (Bubeck et al., 2023; Achiam et al., 2023; Anthropic, 2024; Team et al., 2023). However, the specific algorithms models learn to achieve these capabilities remain largely a mystery to researchers; we do not understand how language models write poetry. Mechanistic interpretability is a field that seeks to address this gap by reverse-engineering trained models from the ground up into variables (features) and the programs (circuits) that process these variables (Olah et al., 2020).

One mechanistic interpretability research direction has focused on understanding toy models in detail. This work has found multi-dimensional representations of inputs such as lattices (Michaud et al., 2024) and circles (Liu et al., 2022; Nanda et al., 2023a), and has successfully reverse-engineered the algorithms that models use to manipulate these representations. A separate direction has identified one-dimensional representations of high level concepts and quantities in large language models (Gurnee & Tegmark, 2023; Marks & Tegmark, 2023; Heinzerling & Inui, 2024; Bricken et al., 2023). These findings have led to the *linear representation hypothesis* (LRH): a hypothesis which has historically claimed both that 1. all representations in pretrained large language models lie along one-dimensional lines, and 2. model states are a simple sparse sum of these representations (Park et al., 2023; Bricken et al., 2023). In this work, we specifically call into question the first part of the LRH.[1]

---

[1]An earlier version of this manuscript sparked discussion in the mechanistic interpretability community on the distinction between non-linear features and multi-dimensional features, and in fact this discussion directly led

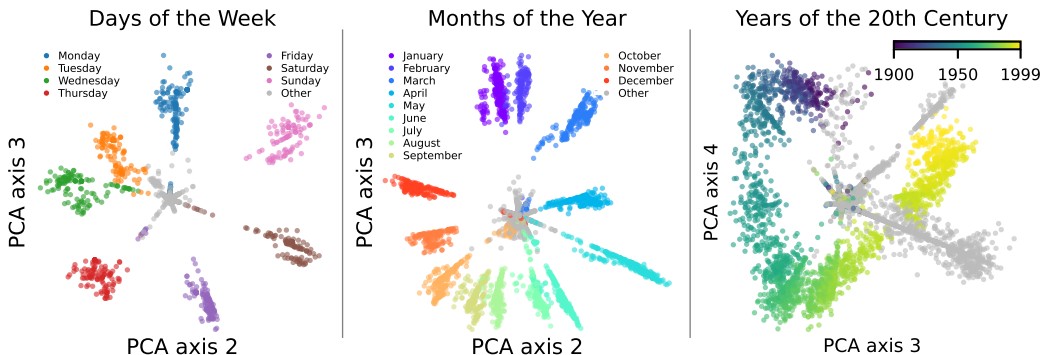

Figure 1: Circular representations of days of the week, months of the year, and years of the 20th century in layer 7 of GPT-2-small colored by the token they fire on. These representations were discovered via clustering SAE dictionary elements, described in Section 4. Points are colored according to the token which created the representation. See Fig. 14 for other axes and Fig. 15 for similar plots for Mistral 7B.

For the most part, these two directions have been disconnected: Yedidia (2023a) and Gould et al. (2023) find intriguing hints of circular language model representations, and Bricken et al. (2023) speculate about the existence of feature manifolds, but these brief results only serve to further emphasize the lack of a unifying and satisfying perspective on the nature of language model features. In this work, we seek to bridge this gap by formalizing, investigating, and systematically searching for multi-dimensional language model features.

## 1.1 CONTRIBUTIONS

1. In Section 3, we generalize the one-dimensional definition of a language model feature to multi-dimensional features and provide an updated multi-dimensional superposition hypothesis to account for these new features.

2. In Section 4, we build on the definitions proposed in Section 3 to develop a theoretically grounded and empirically practical test that uses sparse autoencoders to find irreducible features. Using this test, we identify multi-dimensional representations automatically in GPT-2 and Mistral 7B, including circular representations for the day of the week and month of the year.

3. In Section 5, we show that Mistral 7B and Llama 3 8B use these circular representations when performing modular addition in days of the week and in months of the year. To the best of our knowledge, we are the first to find causal circular representations of concepts in a language model. We additionally find that the model's circular representations respect a continuous notion of time.

## 2 RELATED WORK

**Linear Representations:** Early word embedding methods such as GloVe and Word2vec, although only trained using co-occurrence data, contained directions in their vector spaces corresponding to semantic concepts (Mikolov et al., 2013b; Pennington et al., 2014; Mikolov et al., 2013a). Recent research has found similar evidence of one-dimensional linear representations in sequence models trained only on next token prediction, including Othello board positions (Nanda et al., 2023b; Li et al., 2022), the truth value of assertions (Marks & Tegmark, 2023), and numeric quantities such as longitude, latitude, birth year, and death year (Gurnee & Tegmark, 2023; Heinzerling & Inui, 2024). These results have inspired the linear representation hypothesis (Park et al., 2023; Elhage et al., 2022)

---

to a consensus around the two part LRH we describe above (see (Olah, 2024; Csordás et al., 2024; Mendel, 2024; Kantamneni & Tegmark, 2025)). To clarify, we agree with these discussions, and believe the multi-dimensional features we find are "linear" in the sense that they are contained in a low-dimensional linear subspace, but "non-linear" in the sense that this low-dimensional subspace is not one-dimensional (and this is the sense we mean in the title).

defined above. Jiang et al. (2024) provide theoretical evidence for this hypothesis, assuming a latent (binary) variable-based model of language. Empirically, Bricken et al. (2023) and Cunningham et al. (2023) successfully use sparse autoencoders to break down a model's feature space into an over-complete basis of linear features. These works assume that the number of linear features stored in superposition exceeds the model dimensionality (Elhage et al., 2022).

**Multi-Dimensional Representations:** There has been comparatively little research on multi-dimensional features in language models. Shai et al. (2024) predict and verify that a transformer trained on a hidden Markov model uses a fractal structure to represent the probability of each next token, a clear example of a necessary multi-dimensional feature, but the analysis is restricted to a toy setting. Yedidia (2023a;b) finds that GPT-2 learned position vectors form a helix, which implies a circle when "viewed" from below. Thus, we are not the first to find a circular feature in a language model. However, our work finds circular features that represent latent concepts from text, while the GPT-2 learned position vectors are specific to tokenization, separate from the rest of the model parameters, and causally implicated only due to positional attention masking. Another suggestive result, due to Hanna et al. (2024), is the presence of a U-shape in the representation of numbers between 0 and 100; however, Hanna et al. (2024) find that this representation is not causal, and they only show it exists within a specific prompt distribution. Recent work on dictionary learning (Bricken et al., 2023) has speculated about multi-dimensional *feature manifolds*; our work is similar to this direction and develops the idea of feature manifolds theoretically and empirically. Finally, in a separate direction, Black et al. (2022) argue for interpreting neural networks through the polytopes they split the input space into, and identifies regions of low polytope density as "valid" regions for a potential linear representation.

**Circuits:** Circuits research seeks to identify and understand *circuits*, subsets of a model (usually represented as a directed acyclic graph) that explain specific behaviors (Olah et al., 2020). The base units that form a circuit can be layers, neurons (Olah et al., 2020), or sparse autoencoder features (Marks et al., 2024). In the first circuits-style work, Olah et al. (2020) found line features that were combined into curve detection features in the InceptionV1 image model. More recent work has examined language models, for example the indirect object identification circuit in GPT-2 (Wang et al., 2022). Given the difficulty of designing bespoke experiments, there has been increased research in automated circuit discovery methods (Marks et al., 2024; Conmy et al., 2023; Syed et al., 2023).

**Interpretability for Arithmetic Problems:** Liu et al. (2022) study models trained on modular arithmetic problems $a + b = c \pmod{m}$ and find that models that generalize well have circular representations for $a$ and $b$. Further work by Nanda et al. (2023a) and Zhong et al. (2024) shows that models use these circular representations to compute $c$ via a "clock" algorithm and a separate "pizza" algorithm. These papers are limited to the case of a small model trained only on modular arithmetic. Another direction has studied how large language models perform basic arithmetic, including a circuits level description of the greater-than operation in GPT-2 (Hanna et al., 2024) and addition in GPT-J (Stolfo et al., 2023). These works find that to perform a computation, models copy pertinent information to the token before the computed result and perform the computation in the subsequent MLP layers. Finally, recent work by Gould et al. (2023) investigates language models' ability to increment numbers and finds linear features that fire on tokens equivalent modulo 10.

## 3 DEFINITIONS

This section focuses on hypotheses for how hidden states of language models can be decomposed into sums of functions of the input (features). We focus on $L$ layer transformer models $M$ that take in token input $\mathbf{t} = (t_1, \ldots, t_n)$ from input token distribution $\mathcal{T}$, have hidden states $\mathbf{x}_{1,l}, \ldots, \mathbf{x}_{n,l}$ for layers $l$, and output logit vectors $\mathbf{y}_1, \ldots, \mathbf{y}_n$. Given a set of inputs $T$, we let $X_{i,l}$ be the set of all corresponding $\mathbf{x}_{i,l}$. We write matrices in capital bold, vectors and vector valued functions in lowercase bold, and sets in capital non-bold.

### 3.1 MULTI-DIMENSIONAL FEATURES

**Definition 1** (Feature). We define a $d_f$-*dimensional feature* as a function $\mathbf{f}$ that maps a subset of the input space into $\mathbb{R}^{d_f}$. We say that a feature is *active* on the aforementioned subset.

The input token distribution $\mathcal{T}$ induces a $d_f$-dimensional probability distribution over feature vectors $\mathbf{f}(t)$. As an example, let $n = 1$ (so inputs are single tokens) and consider a feature $\mathbf{f}$ that maps integer tokens to their integer values in $\mathbb{R}^1$. Then $\mathbf{f}$ is a 1-dimensional feature that is active on integer tokens, and $\mathbf{f}(t)$ is the marginal integer occurrence distribution from the token distribution.

How can we differentiate "true" multi-dimensional features from sums of lower dimensional features? We make this distinction by examining the *reducibility* of a potential multi-dimensional feature. That is, $\mathbf{f}$ is a "true" multi-dimensional feature if it cannot be written as the sum of two statistically independent features **and** it cannot be written as the sum of two non-co-occurring features. Formally, we have the following definition:

**Definition 2.** A feature $\mathbf{f}$ is *reducible* into features $\mathbf{a}$ and $\mathbf{b}$ if there exists an affine transformation

$$\mathbf{f} \mapsto \mathbf{R}\mathbf{f} + \mathbf{c} \equiv \begin{pmatrix} \mathbf{a} \\ \mathbf{b} \end{pmatrix} \tag{1}$$

for some orthonormal $d_f \times d_f$ matrix $\mathbf{R}$ and additive constant $\mathbf{c}$, such that the transformed feature probability distribution $p(\mathbf{a}, \mathbf{b})$ satisfies at least one of these conditions:

1. $p$ is *separable*, i.e., factorizable as a product of its marginal distributions:
   $p(\mathbf{a}, \mathbf{b}) = p(\mathbf{a})p(\mathbf{b})$.

2. $p$ is a *mixture*, i.e., a sum of disjoint distributions, one of which is lower dimensional:
   $p(\mathbf{a}, \mathbf{b}) = wp(\mathbf{a})\delta(\boldsymbol{b}) + (1 - w)p(\mathbf{a}, \mathbf{b})$

Here, $p$ is a probability density function that is conditional on the subset of $\mathcal{T}$ that $\boldsymbol{f}$ is active on, $\delta$ is the Dirac delta function, and $0 < w < 1$. By two probability distributions being disjoint, we mean that they have disjoint support (there is no set where both have positive probability measure, or equivalently the two features $\mathbf{a}$ and $\mathbf{b}$ cannot be active at the same time). In Eq. (1), $\mathbf{a}$ is the first $k$ components of the vector $\mathbf{R}\mathbf{f} + \mathbf{c}$ and $\mathbf{b}$ is the remaining $d_f - k$ components. When $p$ is separable or a mixture, we also say that $\mathbf{f}$ is separable or a mixture. We term a feature *irreducible* if it is not reducible, i.e., if no rotation and translation makes it separable or a mixture.

An example of a feature that is a mixture is a one hot encoding along a simplex; an example of a feature that is separable is a normal distribution[2]. In natural language, a mixture might be a one hot encoding of "breed of dog", while a separable distribution might be the "latitude" and "longitude" of location tokens.

In practice, the mixture and separability definitions may not be precisely satisfied. Thus, we soften our definitions to permit degrees of reducibility:

**Definition 3** (Separability Index and $\epsilon$-Mixture Index). Consider a feature $\mathbf{f}$. The **separability index** $S(\mathbf{f})$ measures the minimal mutual information between all possible $\mathbf{a}$ and $\mathbf{b}$ defined in Eq. (1):

$$S(\mathbf{f}) \equiv \min I(\mathbf{a}; \mathbf{b}) \tag{2}$$

where $I$ denotes the mutual information. Smaller values of $S(\mathbf{f})$ mean that $\mathbf{f}$ is more separable.

The $\epsilon$**-mixture index** $M_\epsilon(\mathbf{f})$ tests how often $\mathbf{f}$ can be projected near zero while it is active:

$$M_\epsilon(\mathbf{f}) = \max_{\mathbf{v} \in \mathbb{R}^{d_f}, \, c \in \mathbb{R}} \mathbb{P}_{\boldsymbol{t} \in \mathcal{T}} \left( |\mathbf{v} \cdot \mathbf{f}(\boldsymbol{t}) + c| < \epsilon \sqrt{\mathbb{E}[(\mathbf{v} \cdot \mathbf{f}(\boldsymbol{t}) + c)^2]} \right) \tag{3}$$

Larger values of $M_\epsilon(\mathbf{f})$ mean that $\mathbf{f}$ is more of a mixture.

In Appendix B, we expand on the intuition behind why the separability and $\epsilon$-mixture indices as defined here correspond to weakened versions of Definition 2.

We develop optimization procedures to empirically solve for the separability and $\epsilon$-mixture indices of two dimensional feature distributions. At a high level, the separability procedure iterates over a sweep of rotations and estimates the mutual information between the axes for each angle, while the $\epsilon$-mixture index procedure performs gradient descent to find the $\epsilon$ band that contains the largest possible fraction of the feature distribution. For more details on the implementation of the tests, see Appendix B.2. In Section 4, we apply these empirical tests to real language model feature distributions to find irreducible multi-dimensional features; we show the detailed test results on the "days of the week" cluster in Fig. 2

---

[2]since any multidimensional Gaussian can be rotated to have a diagonal covariance matrix

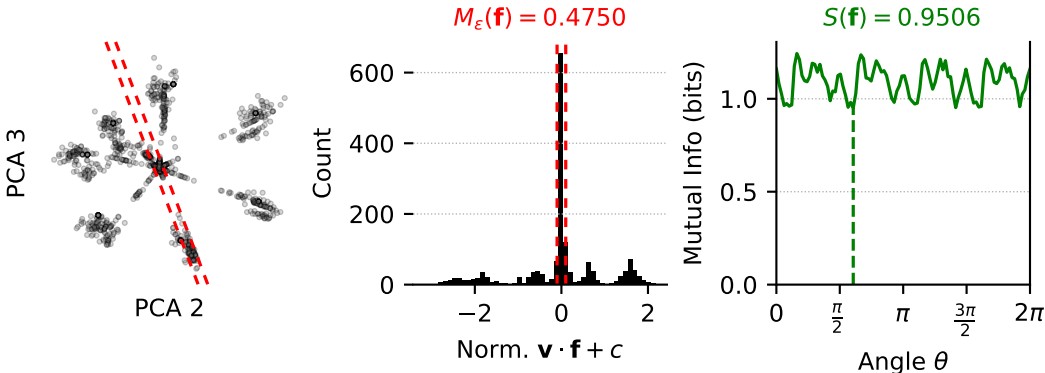

Figure 2: Empirical $\epsilon$-mixture index and separability index for the "days of the week" cluster along PCA components 2 and 3. **Left:** The $\epsilon$ band parameterized by $\mathbf{v}$ and $c$ that the optimization procedure found contained the highest fraction of points. **Mid:** Dot products of points in the feature distribution with the $\epsilon$ band; $M_\epsilon(\boldsymbol{f})$ is the percent of dot products within $\epsilon = 0.1$ of 0. **Right:** Estimated mutual information for different rotations of the space; $S(\boldsymbol{f})$ is the minimum over all rotations. This point cloud has a lower $\epsilon$-mixture index and higher separability index than PCA projections within typical clusters (see Fig. 3), indicating that it is more likely to be an irreducible multi-dimensional feature.

## 3.2 SUPERPOSITION

In this section, we propose an updated *superposition hypothesis* (Elhage et al., 2022) that takes into account multi-dimensional features. First, we restate the original superposition hypothesis:

**Definition 4** ($\delta$-orthogonal matrices). Two matrices $\mathbf{A}_1 \in \mathbb{R}^{d \times d_1}$ and $\mathbf{A}_2 \in \mathbb{R}^{d \times d_2}$ are $\delta$-*orthogonal* if $|\mathbf{x}_1 \cdot \mathbf{x}_2| \leq \delta$ for all unit vectors $\mathbf{x}_1 \in \text{colspace}(\mathbf{A}_1)$ and $\mathbf{x}_2 \in \text{colspace}(\mathbf{A}_2)$.

**Hypothesis 1** (One-Dimensional Superposition Hypothesis, paraphrased from (Elhage et al., 2022)). Hidden states $\mathbf{x}_{i,l}$ are the sum of many ($\gg d$) sparse one-dimensional features $f_i$ and pairwise $\delta$-orthogonal vectors $\mathbf{v}_i$ such that $\mathbf{x}_{i,l}(t) = \sum_i \mathbf{v}_i f_i(t)$. We set $f_i(t)$ to zero when $t$ is outside the domain of $f_i$.

In contrast, our new superposition hypothesis posits independence between irreducible multi-dimensional features instead of unknown levels of independence between one-dimensional features:

**Hypothesis 2** (Multi-Dimensional Superposition Hypothesis, changes underlined). Hidden states $\mathbf{x}_{i,l}$ are the sum of many ($\gg d$) sparse low-dimensional irreducible features $\mathbf{f}_i$ and pairwise $\delta$-orthogonal matrices $\mathbf{V}_i \in \mathbb{R}^{d \times d_{\mathbf{f}_i}}$ such that $\mathbf{x}_{i,l}(t) = \sum_i \underline{\mathbf{V}_i \mathbf{f}_i(t)}$. We set $\mathbf{f}_i(t)$ to zero when $t$ is outside the domain of $\mathbf{f}_i$.

Note that since multi-dimensional features can be written as the sums of projections of lower-dimensional features, our new superposition hypothesis is a stricter version of Hypothesis 1. In the next section, we will explore empirical evidence for our hypothesis, while in Appendix A, we prove upper and lower bounds on the number of $\delta$-almost orthogonal matrices $\mathbf{V}_i$ that can be packed into $d$ dimensional space.

## 4 SPARSE AUTOENCODERS FIND MULTI-DIMENSIONAL FEATURES

In this section, we describe a method to identify multi-dimensional features in language model hidden states using sparse autoencoders (SAEs). Sparse autoencoders (SAEs) deconstruct model hidden states into sparse vector sums from an over-complete basis (Bricken et al., 2023; Cunningham et al., 2023). For hidden states $X_{i,l}$, a one-layer SAE of size $m$ with sparsity penalty $\lambda$ minimizes the following dictionary learning loss (Bricken et al., 2023; Cunningham et al., 2023):

$$\text{DL}(X_{i,l}) = \underset{\mathbf{E} \in \mathbb{R}^{m \times d}, \mathbf{D} \in \mathbb{R}^{d \times m}}{\arg\min} \sum_{\mathbf{x}_{i,l} \in X_{i,l}} \left[ \|\mathbf{x}_{i,l} - \mathbf{D} \cdot \text{ReLU}(\mathbf{E} \cdot \mathbf{x}_{i,l})\|_2^2 + \lambda \|\text{ReLU}(\mathbf{E} \cdot \mathbf{x}_{i,l})\|_0 \right] \quad (4)$$

In practice, the $L_0$ loss on the last term is relaxed to $L_p$ for $0 < p \leq 1$ to make the loss differentiable. We call the $m$ columns of $\mathbf{D}$ (vectors in $\mathbb{R}^d$) **dictionary elements**.

We now argue that SAEs can discover irreducible multi-dimensional features by *clustering* $\mathbf{D}$. We will consider a simple form of clustering: build a complete graph on $\mathbf{D}$ with edge weights equal to the cosine similarity between dictionary elements, prune all edges below a threshold $T$, and then set the clusters equal to the connected components of the graph. If we now consider the spaces spanned by each cluster, they will be approximately $T$-orthogonal by construction, since their basis vectors are all $T$-orthogonal. Now, consider some irreducible two-dimensional feature $\mathbf{f}$; we claim that if the SAE is large enough and $\mathbf{f}$ is active enough such that the SAE can reconstruct $\mathbf{f}$ when $\mathbf{f}$ is active, one of the clusters is likely to be exactly equal to $\mathbf{f}$. If $\mathbf{D}$ includes just two dictionary elements spanning $\mathbf{f}$, then these elements both must have nonzero activations post-ReLU to reconstruct $\mathbf{f}$ (otherwise $\mathbf{f}$ is a mixture). Because of the sparsity penalty in Eq. (4), this two-vector solution to reconstruct $\mathbf{f}$ is disincentivized, so instead the dictionary is likely to learn many elements that span $\mathbf{f}$. These dictionary elements will then have a high cosine similarity, and so the edges between them will not be pruned away during the clustering process; hence, they will be in a cluster.

Thus, we have a way to operationalize Hypothesis 2: clustering $\mathbf{D}$ finds $T$-orthogonal subspaces, and if irreducible multi-dimensional features exist, they are likely to be equal to some of these subspaces. This suggests a natural approach to using sparse autoencoders to search for irreducible multi-dimensional features:

1. Cluster dictionary elements by their pairwise cosine similarity. We use both the simple similarity-based pruning technique described above, as well as spectral clustering; see Appendix F for details, including comments on scalability.

2. For each cluster, run the SAEs on all $\mathbf{x}_{i,l} \in X_{i,l}$ and ablate all dictionary elements not in the cluster. This will give the reconstruction of each $\mathbf{x}_{i,l}$ restricted to the cluster found in step 1 (if no cluster dictionary elements are non-zero for a given point, we ignore the point).

3. Examine the resulting reconstructed activation vectors for irreducible multi-dimensional features. This step can be done manually by visually inspecting the PCA projections for known irreducible multi-dimensional structures (e.g. circles, see Fig. 10) or automatically by passing the PCA projections to the tests for Definition 3.

Pseudocode for this method is in the appendix in Alg. 1. This method succeeds on toy datasets of synthetic irreducible multi-dimensional features; see Appendix D.[3] We apply this method to language models using GPT-2 (Radford et al., 2019) SAEs trained by Bloom (2024) for every layer and Mistral 7B (Jiang et al., 2023) SAEs that we train on layers 8, 16, and 24 (training details in Appendix E).

Strikingly, we reconstruct irreducible multi-dimensional features that are *interpretable* circles: in GPT-2, days, months, and years are arranged circularly in order (see Fig. 1); in Mistral 7B, days and months are arranged circularly in order (see Fig. 15). These plots contain the PCA dimensions that most clearly show circular structure; these best dimensions are usually the second and third because the first PCA dim is an "intensity" direction that manifests as the radius of the circle in Fig. 1 (thus the overall structure for these multi-d features is perhaps best thought of as a cone). See Fig. 14 for all PCA dimensions visualized).

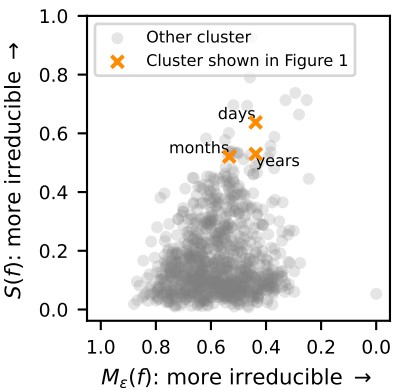

Figure 3: Mixture index and separability index of GPT-2 features. Features from Fig. 1, which we had manually identified, score highly as candidate multidimensional features with these metrics.

For each cluster of GPT-2 SAE features, we take the the reconstructed activations and project them onto PCA components 1-2, 2-3, 3-4, and 4-5 (or fewer if there are fewer features in the cluster) and measure the separability index and $\epsilon$-mixture index of each 2D point cloud as described in Appendix B.2. The mean scores across these planes are a computationally tractable approximation of Definition 3. We plot these mean scores in Fig. 3, and find that the features which we had manually identified in Fig. 1 are among the top scoring features along

---

[3]Code: https://github.com/JoshEngels/MultiDimensionalFeatures

Table 1: Aggregate model accuracy on days of the week and months of the year modular arithmetic tasks. Performance broken down by problem instance in Appendix I.

| Model | Weekdays | Months |
|---|---|---|
| Llama 3 8B | 29 / 49 | 143 / 144 |
| Mistral 7B | 31 / 49 | 125 / 144 |
| GPT-2 | 8 / 49 | 10 / 144 |

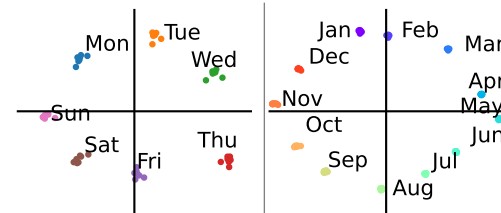

Figure 4: Top two PCA components on the $\alpha$ token. Colors show $\alpha$. **Left:** Layer 30 of Mistral on Weekdays. **Right:** Layer 5 of Llama on Months.

both measures of irreducibility. Thus, our theoretical tests can indeed be used to find interpretable irreducible features. We show the top 20 feature clusters, measured by the product of $(1 - \epsilon$-mixture index ) and separability index, in Appendix G. Out of all 1000 clusters, the Fig. 1 clusters rank $9, 28,$ and 15 by this metric, respectively.[4]

## 5 CIRCULAR REPRESENTATIONS IN LARGE LANGUAGE MODELS

In this section, we examine tasks in which models *use* the multi-dimensional features we discovered in Section 4, thereby providing evidence that these representations are indeed the fundamental unit of computation for some problems. Inspired by prior work studying circular representations in modular arithmetic (Liu et al., 2022), we define two prompts that represent "natural" modular arithmetic tasks:

Weekdays task: *"Let's do some day of the week math. Two days from Monday is"*
Months task: *"Let's do some calendar math. Four months from January is"*

For Weekdays, we range over the 7 days of the week and durations between 1 and 7 days to get 49 prompts. For Months, we range over the 12 months of the year and durations between 1 and 12 months to get 144 prompts. Mistral 7B and Llama 3 8B (AI@Meta, 2024) achieve reasonable performance on the Weekdays task and excellent performance on the Months task (measured by comparing the highest logit valid token against the ground truth answer), as summarized in Table 1. Interestingly, although these problems are equivalent to modular arithmetic problems $\alpha + \beta \equiv ? \pmod{m}$ for $m = 7, 12$, both models get trivial accuracy on plain modular addition prompts, e.g. "5 + 3 (mod 7) ≡". Finally, although GPT-2 has circular representations, it gets trivial accuracy on Weekdays and Months.

To simplify discussion, let $\alpha$ be the day of the week or month of the year token (e.g. "Monday" or "April"), $\beta$ be the duration token (e.g. "four" or "eleven"), and $\gamma$ be the target ground truth token the model should predict, such that (abusing notation) we have $\alpha + \beta = \gamma$. Let the prompts of the task be parameterized by $j$, such that the $j$th prompt asks about $\alpha_j$, $\beta_j$, and $\gamma_j$.

We confirm that Llama 3 8B and Mistral 7B have circular representations of $\alpha$ on this task by examining the PCA projections of hidden states across prompts at various layers on the $\alpha$ token. We plot two of these in Fig. 4 and show all layers in Fig. 18. These plots show circular representations as the highest varying two components in the model's representation of $\alpha$ at many layers.

### 5.1 INTERVENING ON CIRCULAR DAY AND MONTH REPRESENTATIONS

We now experiment with *intervening* on these circular representations. We base our experiments on the common interpretability technique of activation patching, which replaces activations from a "dirty" run of the model with the corresponding activations from a "clean" run (Zhang & Nanda, 2023). Activation patching empirically tests whether a specific model component, position, and/or representation has a causal influence on the model's output. We employ a custom subspace patching

---

[4]We also tried an alternative ranking scheme: we sorted the clusters by separability and irreducibility and set the cluster score equal to the minimum sorted position between the two sorted lists. The Fig. 1 clusters rank $8, 105,$ and 12 by this metric.

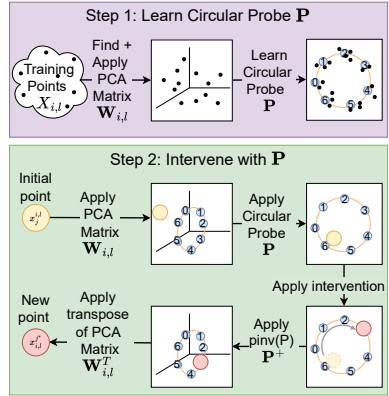
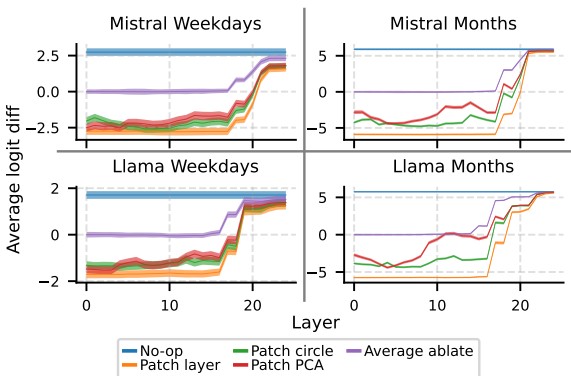

Figure 5: Visual representation of the circular intervention process. **Top:** We learn a circular probe on the PCA projection of a training set. **Bot:** To intervene, we change the circular representation to $\alpha'_j$ and average ablate other dimensions.

Figure 6: Mean and 96% error bars for intervening on the $\alpha$ token across layers using different intervention methods. The circular intervention technique outperforms patching only the top 5 PCA components and leaving the rest unchanged, and almost reaches the upper bound performance of patching the entire layer.

method to allow testing for whether a specific *circular subspace* of a hidden state is sufficient to causally explain model output. Specifically, our patching technique relies on the following steps (visualized in Fig. 5):

1. **Find a subspace with a circle to intervene on**: Using a PCA reduced activation subspace to avoid overfitting, we train a "circular probe" to identify representations which exhibit strong circular patterns. More formally, let $\mathbf{x}_{i,l}^j$ be the hidden state at layer $l$ token position $i$ for prompt $j$. Let $\mathbf{W}_{i,l} \in \mathbb{R}^{k \times d}$ be the matrix consisting of the top $k$ principal component directions of $\mathbf{x}_{i,l}^j$. In our experiments, we set $k = 5$. We learn a linear probe $\mathbf{P} \in \mathbb{R}^{2,k}$ from $\mathbf{W}_{i,l} \cdot X_{i,l}$ to a unit circle in $\alpha$. In other words, if $\texttt{circle}(\alpha) = [\cos(2\pi\alpha/7), \sin(2\pi\alpha/7)]$ for `Weekdays` and $\texttt{circle}(\alpha) = [\cos(2\pi\alpha/12), \sin(2\pi\alpha/12)]$ for `Months`, $\mathbf{P}$ is defined as follows:

$$\mathbf{P} = \underset{\mathbf{P}' \in \mathbb{R}^{2,k}}{\arg\min} \sum_{\mathbf{x}_{i,l}^j} \left\| \mathbf{P}' \cdot \mathbf{W}_{i,l} \cdot \mathbf{x}_{i,l}^j - \texttt{circle}(\alpha) \right\|_2^2 \tag{5}$$

2. **Intervene on the subspace**: Say our initial prompt had $\alpha = \alpha_j$ and we are intervening with $\alpha = \alpha_{j'}$. In this step, we replace the model's projection on the subspace $\mathbf{P} \cdot \mathbf{W}_{i,l}$, which will be close to $\texttt{circle}(\alpha_j)$, with the "clean" point $\texttt{circle}(\alpha_{j'})$. Note that we do not use the hidden state $\mathbf{x}_{i,l}^{j'}$ from the "clean" run, only the "clean" label $\alpha_{j'}$. In practice, other subspaces of $\mathbf{x}_{i,l}^j$ may be used concurrently by the model in "backup" circuits (see e.g. Wang et al. (2022)) to compute the answer, so if we just intervene on the circular subspace the remaining components of the activation may interfere in downstream computations. Thus, to isolate the effect of our intervention, we set the average ablate the portion of the activation not in the intervened subspace. Letting $\overline{\mathbf{x}_{i,l}}$ be the average of $\mathbf{x}_{i,l}^j$ across all prompts indexed by $j$ and $\mathbf{P}^+$ be the pseudoinverse of $\mathbf{P}$, we intervene via the formula

$$\mathbf{x}_{i,l}^{j*} = \overline{\mathbf{x}_{i,l}} + \mathbf{W}_{i,l}^T \mathbf{P}^+ (\texttt{circle}(\alpha_{j'}) - \overline{\mathbf{x}_{i,l}}) \tag{6}$$

We run our patching on all 49 `Weekday` problems and 144 `Month` problems and use as "clean" runs the 6 or 11 other possible values for $\beta$, resulting in a total of $49 * 6$ patching experiments for `Weekdays` and $144 * 11$ patching experiments for `Months`. We also run baselines where we (1) replace the entire subspace corresponding to the first 5 PCA dimensions with the corresponding subspace from the clean run, (2) replace the entire layer with the corresponding layer from the clean run, and (3) replace the entire layer with the average across the task. The metric we use is *average logit difference* across all patching experiments between the original correct token ($\alpha_j$) and the

target token ($\alpha_{j'}$). See Fig. 6 for these interventions on all layers of Mistral 7B and Llama 3 8B on `Weekdays` and `Months`.

The main takeaway from Fig. 6 is that circular subspaces are causally implicated in computing $\gamma$, especially for `Weekdays`. Across all models and tasks, early layer interventions on the circular subspace have almost the same intervention effect as patching the entire layer, and are usually better than patching the top PCA dimensions from the clean problem.

Patching experiments in Appendix J show $\alpha$ is copied to the final token on layers 15 to 17, which is why interventions drop off there. Additionally, while in this section we train a probe on a dataset of prompts, in Section 5.2, we show that intervening on the circle discovered via SAE clustering in Section 4 also works.

To investigate exactly how models use the circular subspace, we perform *off distribution* interventions. We modify Eq. (6) so that instead of intervening on the circumference `circle`($\alpha$), we sweep over a grid of positions $(r, \theta)$ within the circle:

$$\mathbf{x}_{i,l}^{j^*} = \overline{\mathbf{x}_{i,l}} + \mathbf{W}_{i,l}^T \mathbf{P}^+ [r\cos(\theta), r\sin(\theta)]^T - \overline{\mathbf{x}_{i,l}}) \quad (7)$$

We intervene with $r \in [0, 0.1, \ldots, 2], \theta \in [0, 2\pi/100, \ldots, 198\pi/100]$ and record the highest logit $\gamma$ after the forward pass. Fig. 7 displays these results on Mistral layer 5 for $\beta \in [2, 3, 45]$. They imply that Mistral treats the circle as a multi-dimensional representation with $\alpha$ encoded in the angle.

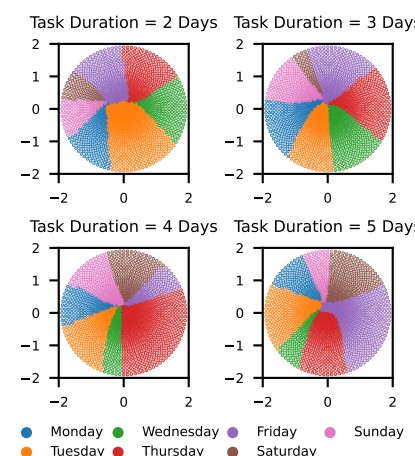

Figure 7: Off distribution interventions on Mistral layer 5 on the `Weekdays` task. The color corresponds to the highest logit $\gamma$ after performing the circular subspace intervention on that point.

## 5.2 INTERVENING WITH THE SAE PLANE

In the last section, we train probes manually on the PCA of the activations to fit a circle. A perhaps more natural approach is intervening with the precise circle we found in Section 4. To determine if this approach is feasible, we first project layer 8 Mistral 7B `Weekdays` activations into the weekdays plane that was discovered by clustering (see Fig. 15; the plane is defined by PCA dimensions 2 and 3 of the cluster). In the rest of this section, we call this plane the SAE plane. In Fig. 19, we find that indeed, the `Weekdays` representations projected into the SAE plane form a circle (see Fig. 19). We thus can fit a circular probe to this 2D plane as in Eq. (5). Similarly, we call this probe the SAE probe.

Because we only have layer 8 clustering results for Mistral, we train an SAE probe only on layer 8. We then evaluate interventions using this SAE probe on layer 8 of Mistral, but also at neighboring layers, since nearby layers should have similar representations (see e.g. Belrose et al. (2023)). We compare to two baselines: 1) training a normal circular probe on the PCA projections of each layer as described in Section 5.1, and 2) training a circular probe only on layer 8 and then evaluating on adjacent layers (in the same way as for the layer 8 SAE probe). We show the results of these methods in Fig. 8.

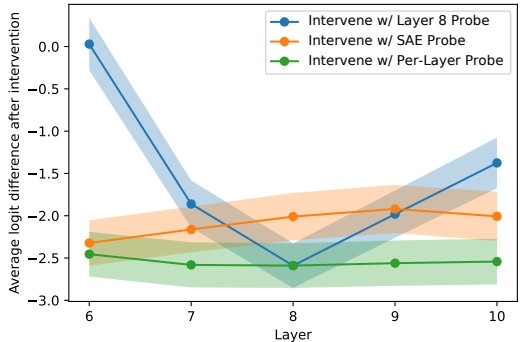

Figure 8: Interventions on the Mistral 7B `Weekdays` task with different methods of determining the probe.

We find that on all layers, using the SAE probe only slightly decreases intervention performance as compared to training a circular probe (from -2.58 to -2.01 average logit difference on layer 8). Even more interestingly, the layer 8 SAE probe is much more robust to layer shifts than the layer 8 circular probe; for example, using the layer 8 circular probe on layer 6 results in an average logit difference of 0.029, whereas using the layer 8 SAE probe results in an average logit difference of

-2.32. This is intriguing evidence that the SAE is perhaps finding more "true" (or at least more robust) features than our circular probing technique.

## 5.3 Continuity of Circular Representations

In past sections, the representations of the interpretable numeric quantities we have discovered have been mostly *discontinuous*; that is, the days of the week and months of the year in Fig. 1 and Fig. 15 are clustered at the vertices of a heptagon and dodecagon, and there is nothing "between" adjacent weekdays or months along the circle. In this section, we will examine the "continuity" of the circular features we have discovered. Although continuity of the representation is not a requirement of Definition 3, it would further decrease the $\epsilon$-mixture index, and would also increase our subjective perception of the circular feature as an intrinsic model feature representing a continuous quantity (time). Thus, we create a synthetic dataset containing the text "[very early/very late] on [Monday/Tuesday/.../Sunday]" and simply plot the projections of the layer 30 activations into the top two PCA components of the activations of [Monday/Tuesday/.../Sunday].

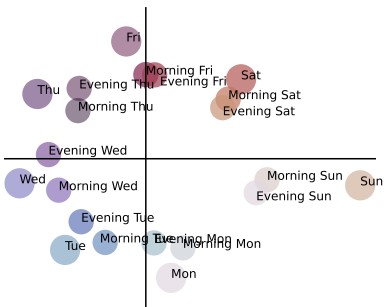

Figure 9: Layer 30 Mistral 7B activations for [morning/evening] on [Monday/Tuesday/.../Sunday], plotted projected into the PCA plane for [Monday/Tuesday/.../Sunday].

The results, shown in Fig. 9, show that Mistral 7B indeed can map intermediate quantities to their expected place in the circle: the very early and very late version of each weekday are more towards the last and the next weekday along the circle, respectively. We show similar results for "[morning/evening] on [Monday/Tuesday/.../Sunday]" in Appendix Fig. 22.

## 6 Discussion

Our work proposes a significant refinement to the simple one-dimensional linear representation hypothesis. While previous work has convincingly shown the existence of one-dimensional features, we find evidence for irreducible multi-dimensional representations, requiring us to generalize the notion of a feature to higher dimensions. Fortunately, we find that existing unsupervised feature extraction methodologies like sparse autoencoders can readily be applied to discover multi-dimensional representations. However, we think our work raises interesting questions about whether individual SAE features are appropriate "mediators" (Mueller et al., 2024) for understanding model computation, if some features are in fact multi-dimensional. Although taking a multi-dimensional representation perspective may be more complicated, we believe that uncovering the true (perhaps multi-dimensional) nature of model representations is necessary for discovering the underlying algorithms that use these representations. Ultimately, our field aims to turn complex circuits in future more-capable models into formally verifiable programs (Tegmark & Omohundro, 2023; Dalrymple et al., 2024), which requires the ground truth "variables" of language models; we believe this work takes an important step towards discovering these variables.

**Limitations:** It is unclear why we did not find more interpretable multi-dimensional features. We are unsure if we are failing to interpret some of the high-scoring multi-dimensional features, if most multi-dimensional features lie in dimensions higher than two, if our clustering technique is not powerful enough to find some features, or if there are truly not that many. Additionally, our definitions for irreducible features (Definition 2) are purely statistical and not intervention based, and also had to be relaxed to hold in practice, resulting in measures that return a possibly subjective "degree" of reducibility (Definition 3). Thus, although this work provides preliminary evidence for the multi-dimensional superposition hypothesis (Hypothesis 2), it is still unclear if this theory provides the best description for the representations models use. Future work might make progress on this question by investigating new techniques for decomposing model representations, exploring higher dimensional representations, or determining conclusively whether models use representations in ways that necessitate the representations are non-linear.

ACKNOWLEDGMENTS

We thank (in alphabetical order) Dowon Baek, Kaivu Hariharan, Vedang Lad, Ziming Liu, and Tony Wang for helpful discussions and suggestions. This work is supported by Erik Otto, Jaan Tallinn, the Rothberg Family Fund for Cognitive Science, the NSF Graduate Research Fellowship (Grant No. 2141064), and IAIFI through NSF grant PHY-2019786.

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

## A  MULTI-DIMENSIONAL FEATURE CAPACITY

The Johnson-Lindenstrauss (JL) Lemma (Johnson & Lindenstrauss, 1984) implies that we can choose $e^{Cd\delta^2}$ pairwise one-dimensional $\delta$-orthogonal vectors to satisfy Hypothesis 1 for some constant $C$, thus allowing us to build the model's feature space with a number of one-dimensional $\delta$-orthogonal features exponential in $d$. We now prove a similar result for low-dimensional projections (the main idea of the proof is to combine $\delta$-orthogonal vectors as guaranteed from the JL lemma):

**Theorem 1.** *For any $d'$ and $\delta$, it is possible to choose $\frac{1}{d_{\max}}e^{C_1(d/d'^2)\delta^2}$ pairwise $\delta$-orthogonal matrices $\mathbf{A}_i \in \mathbb{R}^{n_i \times d'}$ for some constant $C_1$. Furthermore, it is not possible to choose more than $e^{C_2(d - d_{\max}\delta \log(\frac{1}{\delta}))}$ for some constant $C_2$.*

We will first prove a lemma that will help us prove Theorem 1.

**Lemma 1.** *Pick $n$ pairwise $\delta$-orthogonal unit vectors in $\mathbf{v}_1, \dots, \mathbf{v}_n \in \mathbb{R}^d$. Let $\mathbf{y} \in \mathbb{R}^d$ be a unit norm vector that is a linear combination of unit norm vectors $\mathbf{v}_1, \dots, \mathbf{v}_n$ with coefficients $z_1 \dots, z_n \in \mathbb{R}$. We can write $\mathbf{A} = [\mathbf{v}_1, \dots, \mathbf{v}_n]$ and $\mathbf{z} = [z_1, \dots, z_n]^T$, so that we have $\mathbf{y} = \sum_{k=1}^n z_k \mathbf{v}_k = \mathbf{A}\mathbf{z}^T$ with $\|\mathbf{y}\|_2 = 1$. Then,*

$$\left| \sum_{k=1}^n \mathbf{z}_k \right| = \|\mathbf{z}\|_1 \leq \sqrt{\frac{n}{1 - \delta n}}$$

*Proof.* We will first bound the $L_2$ norm of $\mathbf{z}$. If $\sigma_n$ is the minimum singular value of $\mathbf{A}$, then we have via standard singular value inequalities (Higham, 2021)

$$\sigma_n \leq \frac{\|\mathbf{y}\|_2}{\|\mathbf{z}\|_2} \implies \|\mathbf{z}\|_2 \leq \frac{\|\mathbf{y}\|_2}{\sigma_n} = \frac{1}{\sigma_n}$$

Thus we now lower bound $\sigma_n$. The singular values are the square roots of the eigenvalues of the matrix $\mathbf{A}^T\mathbf{A}$, so we now examine $\mathbf{A}^T\mathbf{A}$. Since all elements of $\mathbf{A}$ are unit vectors, the diagonal of $\mathbf{A}^T\mathbf{A}$ is all ones. The off diagonal elements are dot products of pairs of $\delta$-orthogonal vectors, and so are within the range $[-\delta, \delta]$. Then by the Gershgorin circle theorem (Gershgorin, 1931), all eigenvalues $\lambda_i$ of $\mathbf{A}^T\mathbf{A}$ are in the range

$$(1 - \delta(n - 1), 1 + \delta(n - 1))$$

In particular, $\sigma_n^2 = \lambda_n \geq 1 - \delta(n - 1)$, and thus $\sigma_n \geq \sqrt{1 - \delta(n-1)}$. Plugging into our upper bound for $\|\mathbf{z}\|_2$, we have that $\|\mathbf{z}\|_2 \leq 1/\sqrt{1 - \delta(n-1)}$. Finally, the largest $L_1$ for a point on an $n$-hypersphere of radius $r$ is when all dimensions are equal and such a point has magnitude $\sqrt{n}r$, so

$$\|\mathbf{z}\|_1 \leq \sqrt{\frac{n}{1 - \delta(n-1)}} \leq \sqrt{\frac{n}{1 - \delta n}}$$

$\square$

**Theorem 1.** *For any $d'$ and $\delta$, it is possible to choose $\frac{1}{d_{\max}}e^{C_1(d/d'^2)\delta^2}$ pairwise $\delta$-orthogonal matrices $\mathbf{A}_i \in \mathbb{R}^{n_i \times d'}$ for some constant $C_1$. Furthermore, it is not possible to choose more than $e^{C_2(d - d_{\max}\delta \log(\frac{1}{\delta}))}$ for some constant $C_2$.*

*Proof.* By the JL lemma (Johnson & Lindenstrauss, 1984; , https://mathoverflow.net/users/2554/bill johnson), for any $d$ and $\delta$, we can choose $e^{Cd\delta^2}$ $\delta$-orthogonal unit vectors in $\mathbb{R}^d$ indexed as $\mathbf{v}_i$, for some constant $C$. Let $\mathbf{A}_i = [\mathbf{v}_{d_{\max}*i}, \dots, \mathbf{v}_{d_{\max}*i+n_i-1}]$ where each element in the brackets is a column. Then by construction all $\mathbf{A}_i$ are matrices composed of unique $\delta$-orthogonal vectors and there are $\frac{1}{d_{\max}}e^{Cd\delta^2}$ matrices $\mathbf{A}_i$.

Now, consider two of these matrices $\mathbf{A}_i = [\mathbf{v}_1, \dots, \mathbf{v}_{n_i}]$ and $\mathbf{A}_j = [\mathbf{u}_1, \dots, \mathbf{u}_{n_j}]$, $i \neq j$; we will prove that they are $f(\delta)$-orthogonal for some function $f$. Let $\mathbf{y}_i = \sum_{k=1}^{n_i} z_{i,k} \mathbf{v}_k$ be a vector in the colspace of $\mathbf{A}_i$ and $\mathbf{y}_j = \sum_{k=1}^{n_j} z_{j,k} \mathbf{u}_k$ be a vector in the colspace of $\mathbf{A}_j$, such that $\mathbf{y}_i$ and $\mathbf{y}_j$ are unit vectors. To prove $f(\delta)$-orthogonality, we must bound the absolute dot product between $\mathbf{y}_i$ and $\mathbf{y}_j$:

$$\begin{aligned}
|\mathbf{y}_i \cdot \mathbf{y}_j| &= \left| \left( \sum_{k=1}^{n_i} z_{i,k} \mathbf{v}_k \right) \cdot \left( \sum_{k=1}^{n_j} z_{j,k} \mathbf{u}_k \right) \right| \\
&= \left| \sum_{k_1=1}^{n_i} \sum_{k_2=1}^{n_j} (z_{i,k_1} \mathbf{v}_{k_1}) \cdot (z_{j,k_2} \mathbf{u}_{k_2}) \right| \\
&\leq \sum_{k_1=1}^{n_i} \sum_{k_2=1}^{n_j} |z_{i,k_1} z_{j,k_2}| \, |\mathbf{v}_{k_1} \cdot \mathbf{u}_{k_2}| && \text{Triangle Inequality} \\
&\leq \sum_{k_1=1}^{n_i} \sum_{k_2=1}^{n_j} |z_{i,k_1} z_{j,k_2}| \, \delta && \text{All } \mathbf{v}_i, \mathbf{u}_j \text{ are } \delta \text{ orthogonal} \\
&= \delta \sum_{k_1=1}^{n_i} \sum_{k_2=1}^{n_j} |z_{i,k_1} z_{j,k_2}| \\
&= \delta \left| \sum_{k=1}^{n_i} z_{i,k} \right| \left| \sum_{k=1}^{n_j} z_{j,k} \right| && \text{Factoring the product} \\
&\leq \delta \sqrt{\frac{n_i}{1 - \delta n_i}} \sqrt{\frac{n_j}{1 - \delta n_j}} && \text{By Lemma 1} \\
&\leq \frac{\delta d_{\max}}{1 - \delta d_{\max}} && n_i, n_j \leq d_{\max} \text{ by assumption}
\end{aligned}$$

Thus $\mathbf{A}_i$ and $\mathbf{A}_j$ are $f(\delta)$-orthogonal for $f(\delta) = \delta d_{\max}/(1 - \delta d_{\max})$, and so it is possible to choose $\frac{1}{d_{\max}} e^{Cd\delta^2}$ pairwise $f(\delta)$-orthogonal projection matrices. Remapping the variable $\delta$ with $\delta \mapsto f^{-1}(\delta) = \delta/(d_{\max}(1 + \delta))$, we find that it is possible to choose $\frac{1}{d_{\max}} e^{Cd\delta^2/((1+\delta)^2 d_{\max}^2)}$ pairwise $\delta$-orthogonal projection matrices. Because $1 + \delta$ is at most 2 with $\delta \in (0, 1)$, we can further simplify the exponent and find that it is possible to choose $\frac{1}{d_{\max}} e^{C(d/d_{\max}^2)\delta^2/4}$ pairwise $\delta$-orthogonal projection matrices. Absorbing the 4 into the constant $C$ finishes the proof of the lower bound.

For the upper bound, we can proceed much more simply. Consider $k$ pairwise $\delta$-orthogonal matrices $A_i \in \mathbb{R}^{d'}$. Since these matrices are full rank, their column spaces each parameterize a subspace of dimension $d'$, and so by a result from (Alon, 2003) it is possible to choose $e^{Cd'\delta^2 \log(\frac{1}{\delta})}$ almost orthogonal vectors in this subspace. Furthermore, by our definition of $\delta$-orthogonal matrices, all pairs of these vectors between subspaces will be $\delta$-orthogonal. Finally, again by (Alon, 2003) we cannot have more than $e^{Cd\delta^2 \log(\frac{1}{\delta})}$ $\delta$-orthogonal vectors overall, so we have that

$$k e^{C d_{\max} \delta^2 \log(\frac{1}{\delta})} < e^{C d \delta^2 \log(\frac{1}{\delta})}$$

and simplfying,

$$k < e^{C(d - d_{\max}) \delta^2 \log(\frac{1}{\delta})}$$

$\square$

These results imply that models can still represent an exponential number of higher dimensional features. However, there is a large exponential gap between the lower and upper bound we have shown. If the lower bound is reasonably tight, then this would mean that models would be highly incentivized to fit features within the smallest dimensional space possible, suggesting a reason for recent work showing interesting compressed encodings of multi-dimensional features in toy problems (Morwani et al., 2023).

Note that the proof assumes the "worst case" scenario that all of the features are dimension $d_{\max}$, while in practice many of the features may be 1 or low dimensional, so the effect on the capacity of a real model that represents multi-dimensional features is unlikely to be this extreme.

Finally, we note that the dictionary learning literature may have discovered similar results in the past (which we were unaware of), see Foucart & Rauhut (2013).

## B  More on Reducibility

### B.1  Additional Intuition for Definitions

Here, we present some extra intuition and high level ideas for understanding our definitions and the motivation behind them. Roughly, we intend for our definitions in the main text to identify representations in the model that describe an object or concept in a way that fundamentally takes multiple dimensions. We operationalize this as finding a subspace of representations that 1. has basis vectors that "always co-occur" no matter the orientation 2. is not made up of combinations of independent lower-dimensional features.

1. The first condition is met by the mixture part of our definition. The feature in question should be part of an irreducible manifold, and so should "fill" a plane or hyperplane. There shouldn't be any part of the plane where the probability distribution of the feature is concentrated, because this region is then likely part of a lower dimensional feature. The idea of this part of the definition is to capture multi-dimensional objects; if the entire multi-dimensional space is truly being used to represent a high-dimensional object, then the representations for the object should be "spread out" entirely through the space.

2. The second condition is met by the separability part of our definition. This part of the definition is intended to rule out features that co-occur frequently but are fundamentally not describing the same object or concept. For example, latitude and longitude are not a mixture in that they frequently co-occur, but we do not think it is necessarily correct to say they are part of the same multi-dimensional feature because they are independent.

### B.2  Empirical Irreducible Feature Test Details

Our tests for reducibility require the computation of two quantities $S(\mathbf{f})$ for the separability index and $M_\epsilon(\mathbf{f})$ for the $\epsilon$-mixture index. We describe how we compute each index in the following two subsections.

#### B.2.1  Separability Index

We define the separability index in Equation 2 as

$$S(\mathbf{f}) = \min I(\mathbf{a}; \mathbf{b})$$

where the min is over rotations $\mathbf{R}$ used to split $\mathbf{f}' = \mathbf{R}\mathbf{f} + \mathbf{c}$ into $\mathbf{a}$ and $\mathbf{b}$. In two dimensions, the rotation is defined by a single angle, so we can iterate over a grid of 1000 angles and estimate the mutual information between $\mathbf{a}$ and $\mathbf{b}$ for each angle. We first normalize $\mathbf{f}$ by subtracting off the mean and then dividing by the root mean squared norm of $\mathbf{f}$ (and multiplying by $\sqrt{2}$ since the toy datasets are in two dimensions). To estimate the mutual information, we first clip the data $\mathbf{f}$ to a 6 by 6 square centered on the origin. We then bin the points into a 40 by 40 grid, to produce a discrete distribution $p(a, b)$. After computing the marginals $p(a)$ and $p(b)$ by summing the distribution over each axis, we obtain the mutual information via the formula

$$I(\mathbf{a}; \mathbf{b}) = \sum_{a,b} p(a,b) \log \frac{p(a,b)}{p(a)p(b)} \tag{8}$$

#### B.2.2  $\epsilon$-Mixture Index

We define the $\epsilon$-mixture index in Equation 3 as

$$M_\epsilon(\mathbf{f}) = \max_{\mathbf{v} \in \mathbb{R}^{d_f},\, c \in \mathbb{R}} \mathbb{P}\left(|\mathbf{v} \cdot \mathbf{f} + c| < \epsilon\sqrt{\mathbb{E}[(\mathbf{v} \cdot \mathbf{f} + c)^2]}\right)$$

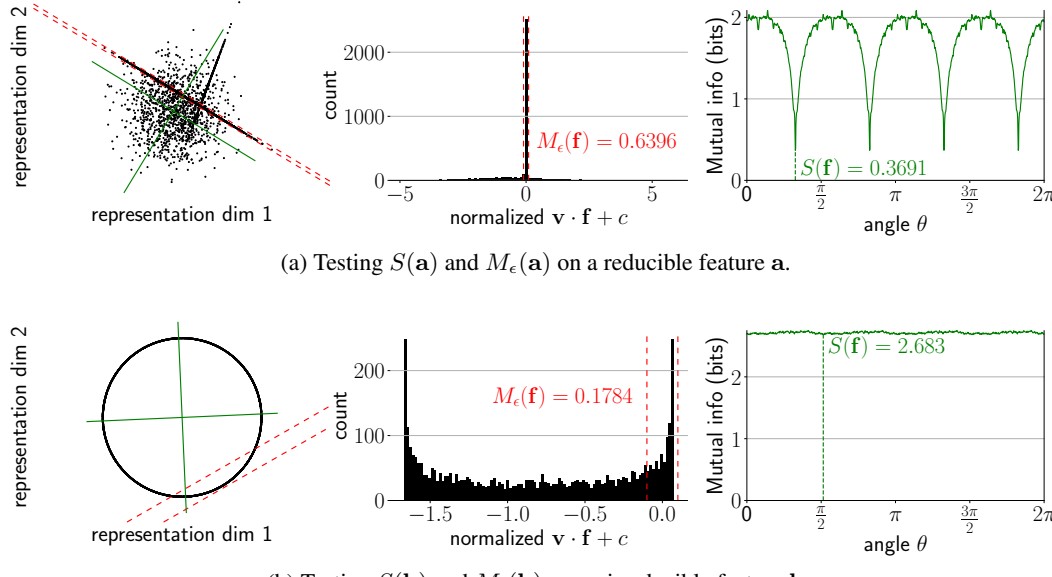

(a) Testing $S(\mathbf{a})$ and $M_\epsilon(\mathbf{a})$ on a reducible feature $\mathbf{a}$.

(b) Testing $S(\mathbf{b})$ and $M_\epsilon(\mathbf{b})$ on an irreducible feature $\mathbf{b}$

Figure 10: Testing irreducibility of synthetic features. **Left in each subfigure:** Distributions of $\mathbf{x}$. For feature $\mathbf{a}$, 63.96% lies within the narrow dotted lines, indicating the feature is likely a mixture. For feature $\mathbf{b}$, 17.84% lies within the wide lines, indicating the feature is unlikely to be a mixture. The green cross indicates the angle $\theta$ that minimizes mutual information. **Middle in each subfigure:** Histograms of the distribution of $\mathbf{v} \cdot \mathbf{x}$ with red lines indicating a $2\epsilon$-wide region. **Right in each subfigure:** Mutual information between $\mathbf{a}$ and $\mathbf{b}$ as a function of the rotation angle $\theta$ of matrix $\mathbf{R}$. Feature $\mathbf{b}$ has a large minimum mutual information so is unlikely to be separable; feature $\mathbf{a}$ has a medium value of minimum mutual information of about 0.37 bits.

The challenge with computing $M_\epsilon(\mathbf{f})$ is to compute the maximum. We opted to maximize via gradient descent; and we guaranteed differentiability by softening the inequality $<$ with a sigmoid,

$$M_{\epsilon,T}(\mathbf{f}, \mathbf{v}, c) = \mathbb{E}\left( \sigma\left( \frac{1}{T}\left( \epsilon - \frac{|\mathbf{v} \cdot \mathbf{f} + c|}{\sqrt{\mathbb{E}[(\mathbf{v} \cdot \mathbf{f} + c)^2]}} \right) \right) \right) \tag{9}$$

where $T$ is a temperature, which we linearly decay from 1 to 0 throughout training. We optimize for $\mathbf{v}$ and $c$ using this loss $M_{\epsilon,T}(\mathbf{f}, \mathbf{v}, c)$ using full batch gradient descent over 10000 steps with learning rate 0.1. With the solution $(\mathbf{v}^*, c^*)$, the final value of $M_{\epsilon,T=0}(\mathbf{f}, \mathbf{v}^*, c^*)$ is then our estimate of $M_\epsilon(\mathbf{f})$.

We also run the irreducibility tests on additional synthetic feature distributions in Fig. 11a and Fig. 11b.

## C  ALTERNATIVE DEFINITIONS

In this section, we present an alternative definition of a reducible feature that we considered during our work. This chiefly deals with multi-dimensional features from the angle of *computational* reducibility as opposed to *statistical* reducibility. In other words, this definition considers whether representations of features on a specific set of tasks can be split up without changing the accuracy of the task. This captures an interesting (and important) aspect of feature reducibility, but because it requires a specific set of prompts (as opposed to allowing unsupervised discovery) we chose not to use it as our main definition.

Our alternative definitions consider *representation spaces* that are possibly multi-dimensional, and defines these spaces through whether they can completely explain a function $h$ on the output logits. We consider a *group theoretic* approach to irreducible representations, via whether computation involving multiple group elements can be decomposed.

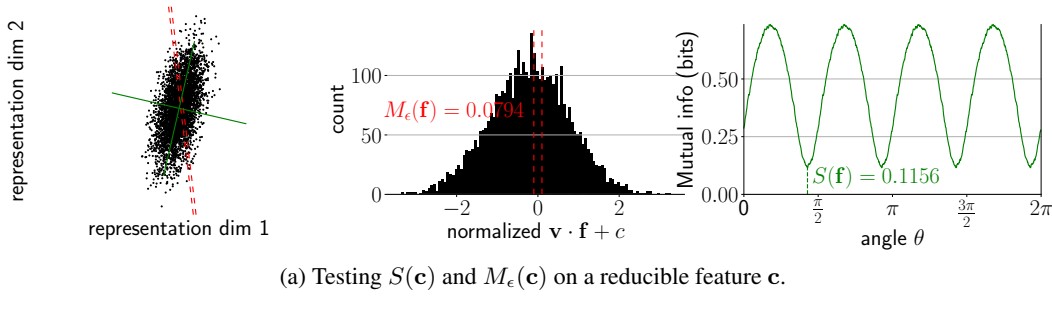

(a) Testing $S(\mathbf{c})$ and $M_\epsilon(\mathbf{c})$ on a reducible feature $\mathbf{c}$.

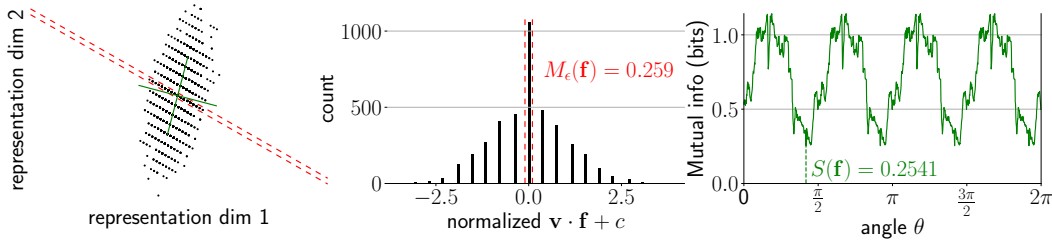

(b) Testing $S(\mathbf{d})$ and $M_\epsilon(\mathbf{d})$ on an irreducible feature $\mathbf{d}$

Figure 11: Testing irreducibility of synthetic features. **Left in each subfigure:** Distributions of $\mathbf{x}$. For feature $\mathbf{c}$, 7.94% lies within the narrow dotted lines, indicating the feature is unlikely to be a mixture. For feature $\mathbf{d}$, 25.90% lies within the wide lines, indicating the feature is likely a mixture. The green cross indicates the angle $\theta$ that minimizes mutual information. **Middle in each subfigure:** Histograms of the distribution of $\mathbf{v} \cdot \mathbf{x}$ with red lines indicating a $2\epsilon$-wide region. **Right in each subfigure:** Mutual information between $\mathbf{a}$ and $\mathbf{b}$ as a function of the rotation angle $\theta$ of matrix $\mathbf{R}$. Both features have a small ($< 0.5$ bits) minimum mutual information and so are likely separable.

## C.1 Alternative Definition: Interventions and Representation Spaces

Assume that we restrict the input set of prompts $T = \{\mathbf{t}^j\}$ to some subset of prompts and that we have some evaluation function $h$ that maps from the output logit distribution of $M$ to a real number. For example, for the `Weekdays` problems, $T$ is the set of 49 prompts and $h$ could be the $\arg\max$ over the days of week logits. Abusing notation, we let $M$ also be the function from the layer we are intervening on; this is always clear from context. Then we can define a representation space of $\mathbf{x}_{i,l}^j$ as a subspace in which interventions always work:

**Definition 5** (Representation Space). Given a prompt set $T = \{\mathbf{t}^j\}$, a rank-$r$ dimensional *representation space* of intermediate value $\mathbf{x}_{i,l}^j$ is a rank $r$ projection matrix $P$ such that for all $j, j'$,

$$h(M((I - P)\mathbf{x}_{i,l}^j + P\mathbf{x}_{i,l}^{j'})) = h(M(\mathbf{x}_{i,l}^{j'})).$$

Note that it immediately follows that the rank $d$ dimensional matrix $I_d$ is trivially a rank $d$ representation space for all prompt sets $T$.

**Definition 6** (Minimality). A representation space $P$ of rank $r$ is *minimal* if there does not exist a lower rank representation space.

A minimal representation with rank $> 1$ is a *multi-dimensional representation*.

**Definition 7** (Alternative Reducibility). A representation space $P$ of rank $r$ is *reducible* if there are orthonormal representation spaces $P_1$ and $P_2$ (such that $P_1 + P_2 = P$, $P_1 P_2 = 0$) where

$$h(M(P_1\mathbf{x}_{i,l}^j) + M(P_2\mathbf{x}_{i,l}^j)) = h(M(P_1\mathbf{x}_{i,l}^j + P_2\mathbf{x}_{i,l}^j))$$

for all $j, j'$.

Suppose $T$, $h$ and $M$ define the multiplication of two elements in a finite group $G$ of order $n$. Then if we interpret the embedding vectors as the group representations, our definition of reducibility implies to the standard group-theoretic definition of irreducibility — specifically, reducibility into a tensor product representation.

# D  TOY CASE OF TRAINING SAEs ON CIRCLES

To explore how SAEs behave when reconstructing irreducible features of dimension $d_f > 1$, we perform experiments with the following toy setup. Inspired by the circular representations of integers that networks learn when trained on modular addition (Nanda et al., 2023a; Liu et al., 2022), we create synthetic datasets of activations containing multiple features which are each 2d irreducible circles.

First however, consider activations for a single circle – points uniformly distributed on the unit circle in $\mathbb{R}^2$. We train SAEs on this data with encoder $\texttt{Enc}(\mathbf{x}) = \texttt{ReLU}(\mathbf{W_e}(\mathbf{x} - \mathbf{b_d}) + \mathbf{b_e})$ and decoder $\texttt{Dec}(\mathbf{f}) = \mathbf{W_d}\mathbf{f} + \mathbf{b_d}$. We train SAEs with $m = 2$ and $m = 10$ with the Adam optimizer and a learning rate of $10^{-3}$, sparsity penalty $\lambda = 0.1$, for 20,000 steps, and a warmup of 1000 steps. In Fig. 12 we show the dictionary elements of these SAEs. When $m = 2$, the SAE must use both SAE features on each input point, and uses $\mathbf{d_b}$ to shift the reconstructed circle so it is centered at the origin. When $m = 10$, the SAE learns $\mathbf{d_b} \approx 0$ and the features spread out across the circle, arranged close together, and only a subset are active on each input.

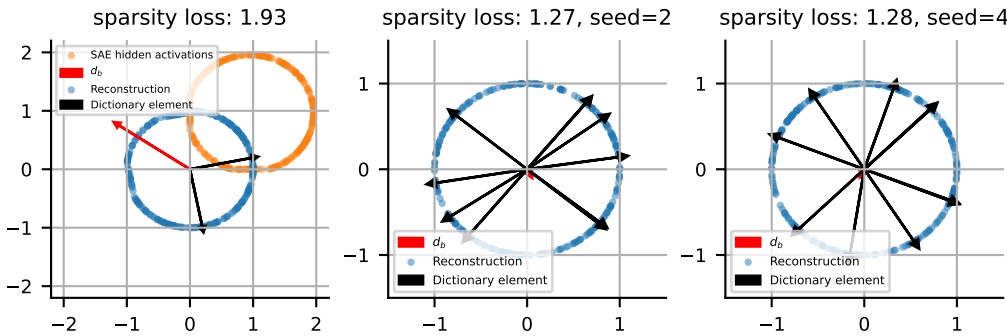

Figure 12: SAEs trained to reconstruct a single 2d circle with $m = 2$ (left) and $m = 10$ (middle and right) dictionary elements. When there are several SAE features, there is no natural/canonical choice of feature directions, and the dictionary elements spread out across the circle.

We now consider synthetic activations with multiple circular features. Our data consists of points in $\mathbb{R}^{10}$, where we choose two orthogonal planes spanned by $(\mathbf{e_1}, \mathbf{e_2})$ and $(\mathbf{e_3}, \mathbf{e_4})$, respectively. With probability one half a points is sampled uniformly on the unit circle in the $\mathbf{e_1}$-$\mathbf{e_2}$ plane, otherwise the point will be sampled uniformly on the unit circle in the $\mathbf{e_3}$-$\mathbf{e_4}$ plane. We train SAEs with $m = 64$ on this data with the same hyperparameters as the single-circle case.

We now apply the procedure described in Section 4 to see if we can automatically rediscover these circles. Encouragingly, we first find that the alive SAE features align almost exactly with either the $\mathbf{e_1}$-$\mathbf{e_2}$ or the $\mathbf{e_3}$-$\mathbf{e_4}$ plane. When we apply spectral clustering with $\texttt{n\_clusters} = 2$ to the features with the pairwise angular similarities between dictionary elements as the similarity matrix (Fig. 13, left), the two clusters correspond exactly to the features which span each plane. As described in Section 4, given a cluster of dictionary elements $S \subset \{1, \ldots, m\}$, we run a large set of activations through the SAE, then filter out samples which don't activate any element in $S$. For samples which do activate an element of $S$, reconstruct the activation while setting all SAE features not in $S$ to have a hidden activation of zero. If some collection of SAE features together represent some irreducible feature, we want to remove all other features from the activation vector, and so we only allow SAE features in the collection to participate in reconstructing the input activation. We find that this procedure almost exactly recovers the original two circles, which encouraged us to apply this method for discovering the features shown in Fig. 1 and Fig. 15.

# E  TRAINING MISTRAL SAEs

Our Mistral 7B (Jiang et al., 2023) sparse autoencoders (SAEs) are trained on over one billion tokens from a subset of the Pile (Gao et al., 2020) and Alpaca (Peng et al., 2023) datasets. We train our SAEs

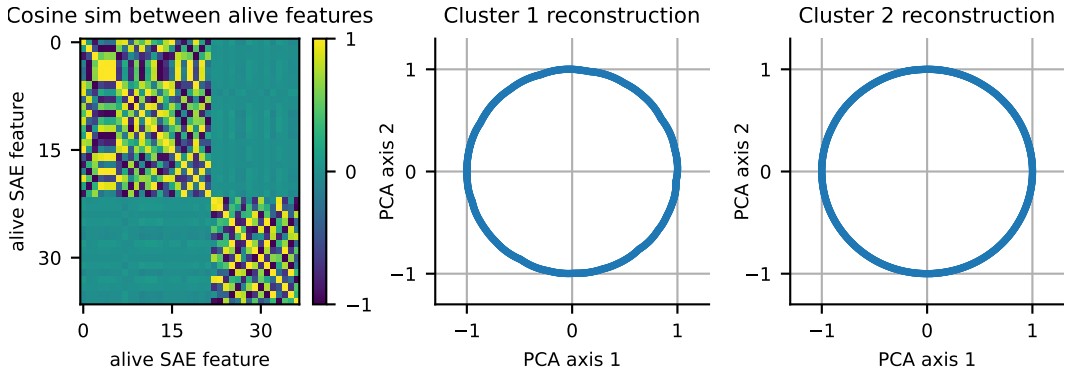

Figure 13: Automatic discovery of synthetic circular features by clustering SAE dictionary elements.

on layers 8, 16, and 24 out of 32 total layers to maximize coverage of the model's representations. We use a $16\times$ expansion factor, yielding a total of 65536 dictionary elements for each SAE.

To train our SAEs, we use an $L_p$ sparsity penalty for $p = 1/2$ with sparsity coefficient $\lambda = 0.012$. Before an SAE forward pass, we normalize our activation vectors to have norm $\sqrt{d_{model}} = 64$ in the case of Mistral. We do not apply a pre-encoder bias. We use an AdamW optimizer with weight decay $10^{-3}$ and learning rate 0.0002 with a linear warm up. We apply dead feature resampling (Bricken et al., 2023) five times over the course of training to converge on SAEs with around 1000 dead features.

## F GPT-2 AND MISTRAL 7B DICTIONARY ELEMENT CLUSTERING

In this section, we first present pseudocode in Alg. 1 for the overall high level technique that finds multi-dimensional features and that uses clustering as a subroutine. We then provide the specific clustering algorithm implementations we use for GPT-2 and Mistral.

---

**Algorithm 1:** High Level Clustering Approach For Finding Multi-D Features

---

**Input:** Dictionary elements $D$, activation vectors $X_{i,l}$, SAE
**Output:** Irreducible multi-dimensional features
$S_{i,j} \leftarrow$ CosineSim$(D_i, D_j)$;
$clusters \leftarrow$ Cluster$(S)$;
$reconstructions \leftarrow \{\}$;
**for** $cluster$ in $clusters$ **do**
    $R_{cluster} \leftarrow$ ids of dictionary elements in $cluster$;
    **for** $\boldsymbol{x}_{i,l}$ in $X_{i,l}$ **do**
        $encoding \leftarrow$ ReLU$(E \cdot \boldsymbol{x}_{i,l})$;
        **if** $\max(encoding[R_{cluster}]) > 0$ **then**
            $r \leftarrow D[:, R_{cluster}] \cdot encoding$;
            $reconstructions \leftarrow reconstructions \cup \{r\}$;
        **end**
    **end**
**end**
$features \leftarrow \{\}$;
**for** $R$ in $reconstructions$ **do**
    $proj \leftarrow$ PCA$(R)$;
    **if** TestIrreducible$(proj)$ **then**
        Add $proj$ to $features$;
    **end**
**end**
**return** $features$

---

## F.1 GPT-2-SMALL METHODS AND RESULTS

For GPT-2-small, we perform spectral clustering on the roughly 25k layer 7 SAE features from (Bloom, 2024), using pairwise angular similarities between dictionary elements as the similarity matrix. We use n_clusters = 1000 and manually looked at roughly 500 of these clusters. For each cluster, we looked at projections onto principal components 1-4 of the reconstructed activations for these clusters. In Fig. 14, we show projections for the most interesting clusters we identified, which appear to be circular representations of days of the week, months of the year, and years of the 20th century.

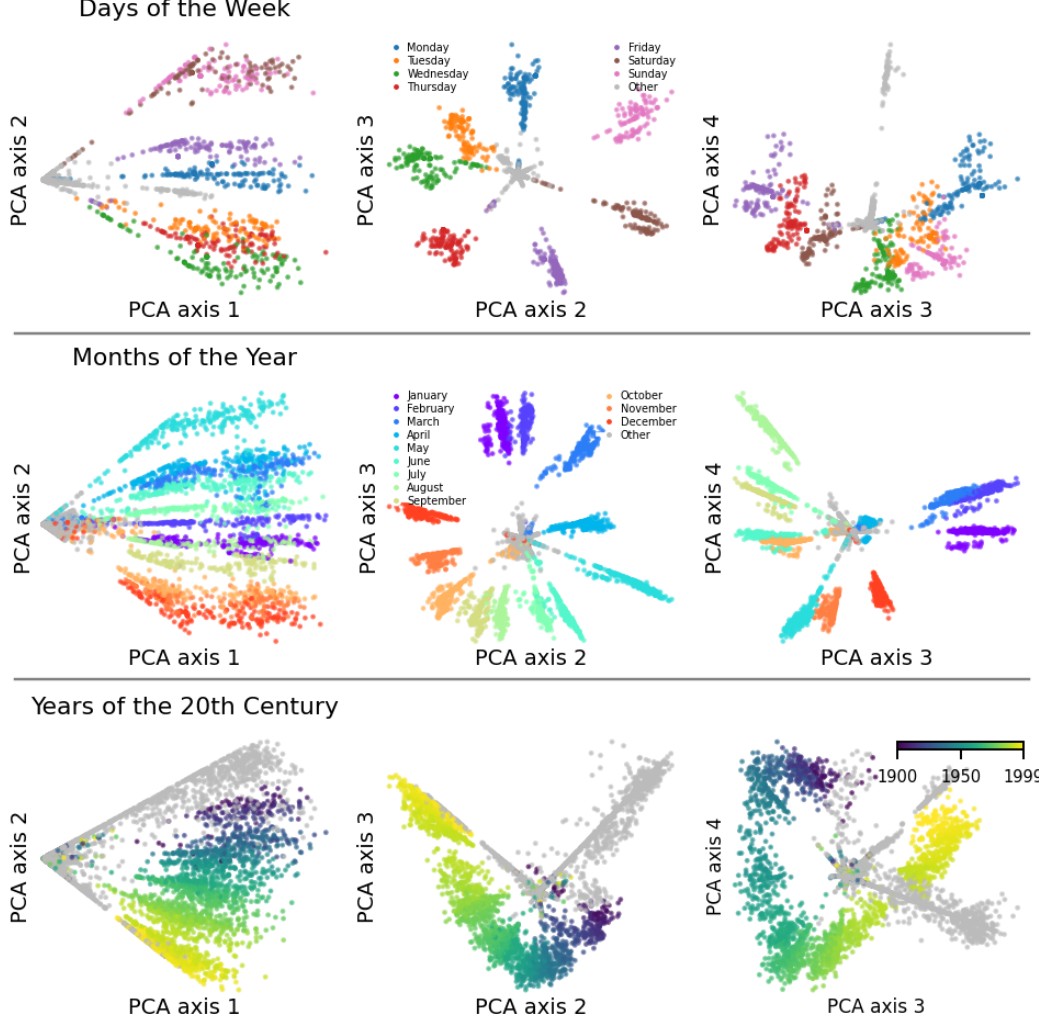

Figure 14: Projections of days of week, months of year, and years of the 20th century representations onto top four principal components, showing additional dimensions of the representations than Fig. 1.

## F.2 MISTRAL 7B METHODS AND RESULTS

For Mistral 7B, our SAEs have 65536 dictionary elements and we found it difficult to run spectral clustering on all of these at once. We therefore develop a simple graph based clustering algorithm that we run on Mistral 7B SAEs:

1. Create a graph $G$ out of the dictionary elements by adding directed edges from each dictionary element to its $k$ closest dictionary elements by cosine similarity. We use $k = 2$.

2. Make the graph undirected by turning every directed edge into an undirected edge.

3. Prune edges with cosine similarity less than a threshold value $\tau$. We use $\tau = 0.5$.

4. Return the connected components as clusters.

We run this algorithm on the Mistral 7B layer $8$ SAE ($2^{16}$ dictionary elements) and find roughly $2700$ clusters containing between $2$ and $1000$ elements. We manually inspected roughly $2000$ of these. From these, we re-discover circular representations of days of the week and months of the year, shown in Fig. 15. However, we did not find other obviously interesting and clearly irreducible features.

We also investigate the sensitivity of this method to $\tau$ and $k$ by varying $\tau$ and $k$ and showing the max Jaccard similarity between any of the resulting clusters and the days of the week cluster we show in Fig. 15. We show the results in Fig. 16, where we find that varying $k$ has minimal effect, while varying $\tau$ shows 3 regimes: small $\tau$ causes all features to group in one cluster, so the days of the week cluster is not found; medium $\tau$ causes the days of the week cluster to become identifiable; large $\tau$ causes all features to be divided into their own clusters.

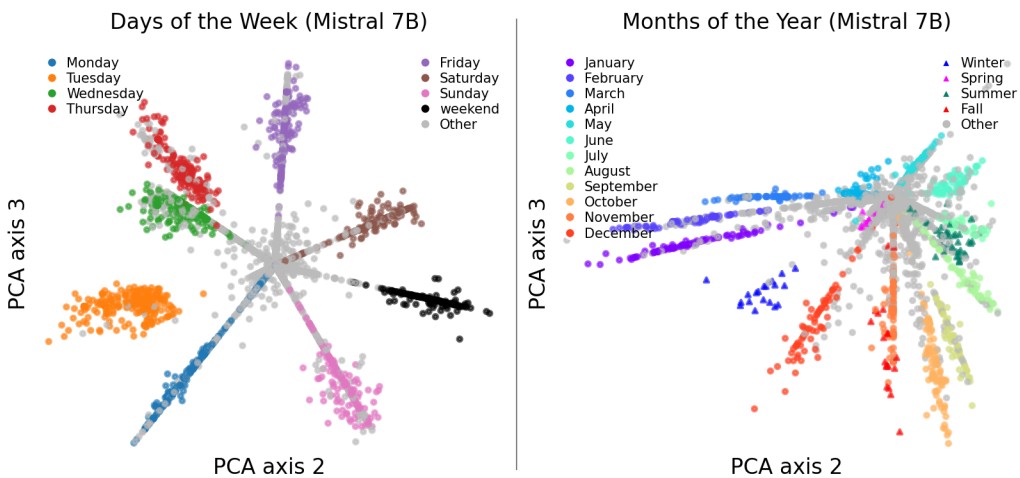

Figure 15: Circular representations of days of the week and months of the year which we discover with our unsupervised SAE clustering method in Mistral 7B. Unlike similar features in GPT-2, we also find an additional "weekend" representation in between Saturday and Sunday representations (left) and additional representations of seasons among the months (right). For instance, "winter" tokens activate a region of the circle in between the representation of January and December.

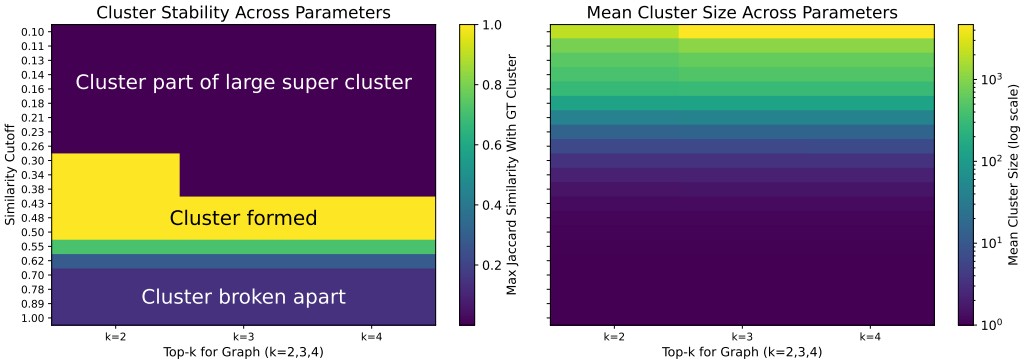

Figure 16: Hyperparameter regimes where the days of the week cluster exists. The cluster exists in the regime between all features clumping together and all features being in their own cluster; this regime seems reasonably stable.

As future work, we think it would be exciting to develop better clustering techniques for SAE features. Our graph based clustering technique could likely be improved by more recent efficient and high-quality graph based clustering techniques, e.g. hierarchical agglomerate clustering with single-linkage (Lattanzi et al., 2020). Additionally, we believe we would see a large improvement by setting edge weights to be a combination of both the cosine and Jaccard similarity of the dictionary elements, e.g. max(cosine, Jaccard).

## G  OTHER DISCOVERED CLUSTERS

In Fig. 17, we plot the top 11 ranked clusters by the product of a) the measured separability index and b) one minus the measured $\epsilon$-mixture index with $\epsilon = 0.1$ (this is just one of many possible ways to get an ordered ranking from a two-parameter score). We color by both the current token (which results in clear patterns for all tokens) and the next token (to see if we find belief states as found by Shai et al. (2024) in toy transformers). We note that weekdays are ranked 9 and so are shown in the plot. Additionally, the next token patterns of the 'such' cluster and the 'B' cluster do seem to display some clustering independently of the the current token pattern, which might lend the belief state hypothesis some support.

## H  FURTHER EXPERIMENT DETAILS

### H.1  ASSETS INFORMATION

We use the following open source models for our experiments: Llama 3 8B (AI@Meta, 2024) (custom Llama 3 license `https://llama.meta.com/llama3/license/`), Mistral 7B (Jiang et al., 2023) (released under the Apache 2 License), and GPT-2 (Radford et al., 2019) (modified MIT license, see `https://github.com/openai/gpt-2/blob/master/LICENSE`).

### H.2  MACHINE INFORMATION

Intervention experiments were run on two V100 GPUs using less than 64 GB of CPU RAM; all experiments can be reproduced from our open source repository in less than a day with this configuration. We use the TransformerLens library (Nanda & Bloom, 2022) for intervention experiments. $\epsilon$-mixture index measurements on toy datasets took about one minute each, on 8GB of CPU RAM. EVR experiments take seconds on 8GB of CPU RAM and are dominated by time taken to human-interpret the RGB plots.

GPT-2 SAE clustering and plotting was run on a cluster of heterogeneous hardware. Spectral clustering and computing reconstructions + plotting was done on CPUs only. We made reconstruction plots for 500 clusters, with each taking less than 10 minutes. Mistral 7B SAE reconstruction plots were made on the same cluster. We made roughly 2000 reconstruction plots for Mistral 7B (and manually inspected each), with each taking less than 20 minutes to generate. Jobs were allocated 64GB of memory each.

Mistral SAE training was run on a single V100 GPU. Initially caching activations from Mistral 7B on one billion tokens took approximately 60 hours. Training the SAEs on the saved activations took another 36 hours.

### H.3  ERROR BAR CALCULATION

In Fig. 6 we report 96% error bars for all intervention methods. To compute these error bars, we loop over all intervention methods and all layers and compute a confidence interval for each (method, layer) pair across all prompts. Assuming normally distributed errors, we compute error bars with the following standard formula:

$$EB = \mu \pm z * SE$$

where $\mu$ is the sample mean, $z$ is the z score (slightly larger than 2 for 96% error bars), and $SE$ is the standard error (the standard deviation divided by the square root of the number of samples). We use standard Python functions to compute this value.

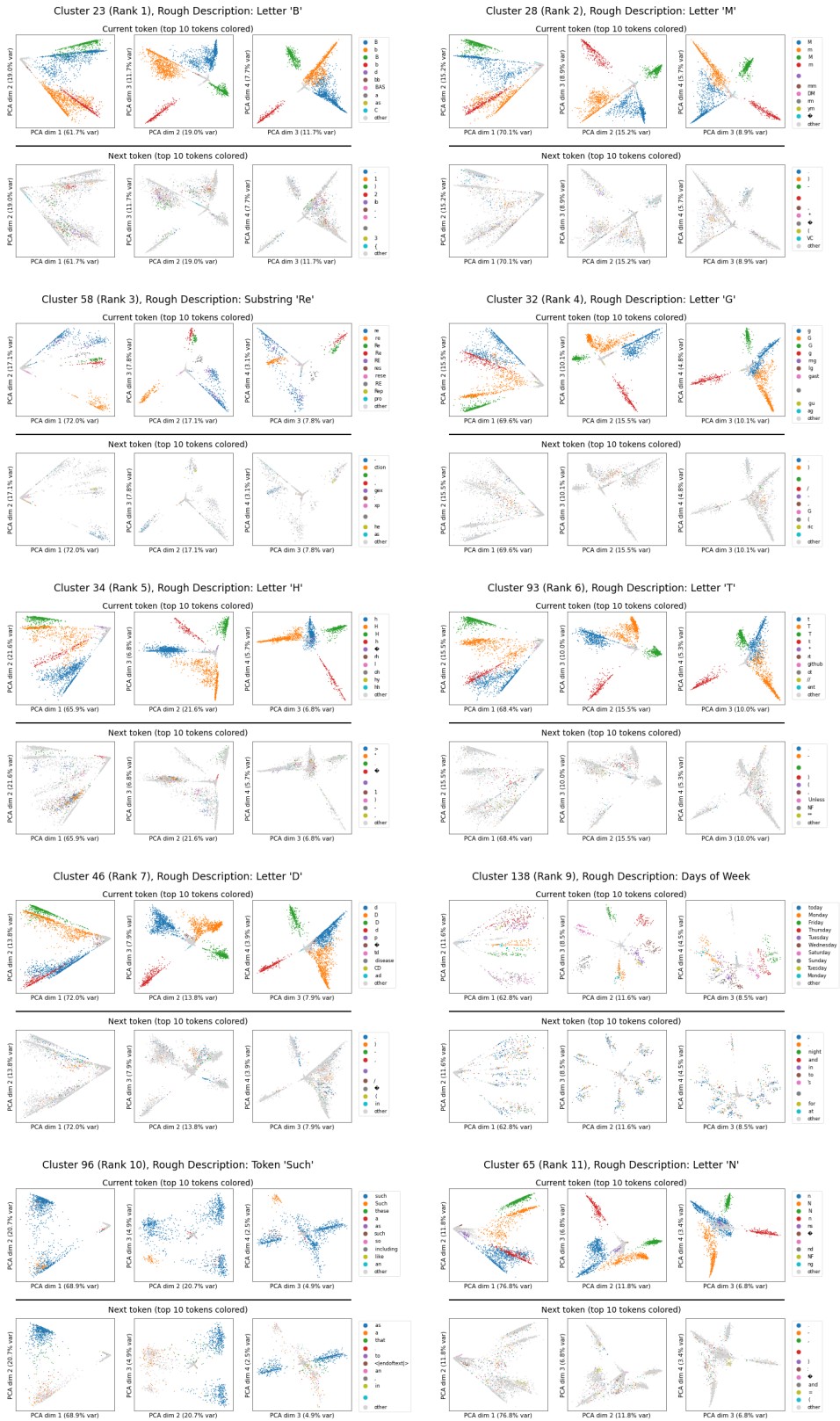

Figure 17: Top 10 GPT-2 clusters by Mixture and Separability Index.

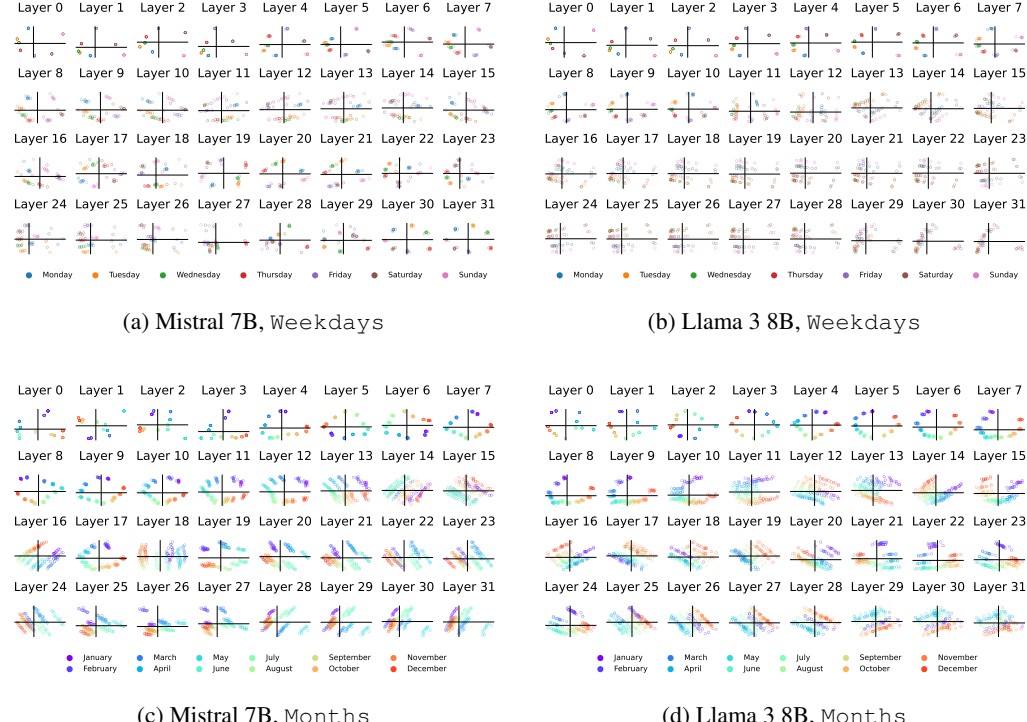

(a) Mistral 7B, `Weekdays`

(b) Llama 3 8B, `Weekdays`

(c) Mistral 7B, `Months`

(d) Llama 3 8B, `Months`

Figure 18: Projections onto the top two PCA dimensions of model hidden states on the $\alpha$ token show that circular representations of $\alpha$ are present in various layers.

The reason that the `Months` error bars are smaller than the `Weekdays` error bars is because there are more `Months` prompts: there are $12 * 12 * 11 = 1584$ intervention effect values, rather than $7 * 7 * 6 = 294$ intervention effect values.

## I MORE WEEKDAYS AND MONTHS PLOTS AND DETAILS

### I.0.1 BASIC PLOTS

We show the results of Mistral 7B and Llama 3 8B on all individual instances of `Weekdays` that at least one of the models get wrong in Table 2 and present a similar table for `Months` in Table 3.

We show projections onto the top two PCA directions for both Mistral 7B and Llama 3 8B in Fig. 18 on the hidden layers on top of the $\alpha$ token, colored by $\alpha$. These are similar plots to Fig. 4, except they are on all layers. The circular structure in $\alpha$ is visible on many—but not all—layers. Much of the linear structure visible is due to $\beta$.

### I.0.2 INTERVENING WITH THE SAE PROBE

We show the results of projecting Mistral `Weekdays` representations into the plane discovered by clustering SAE features in Fig. 19. The result is clearly circular.

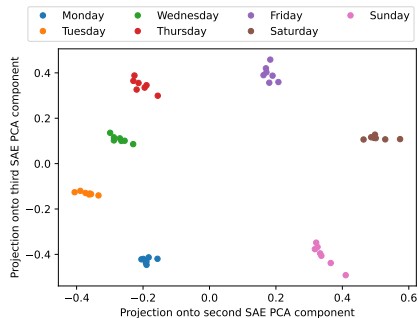

Figure 19: Projections of Mistral 7B `Weekdays` task activations at layer 8 into the plane discovered by clustering layer 8 SAE features.

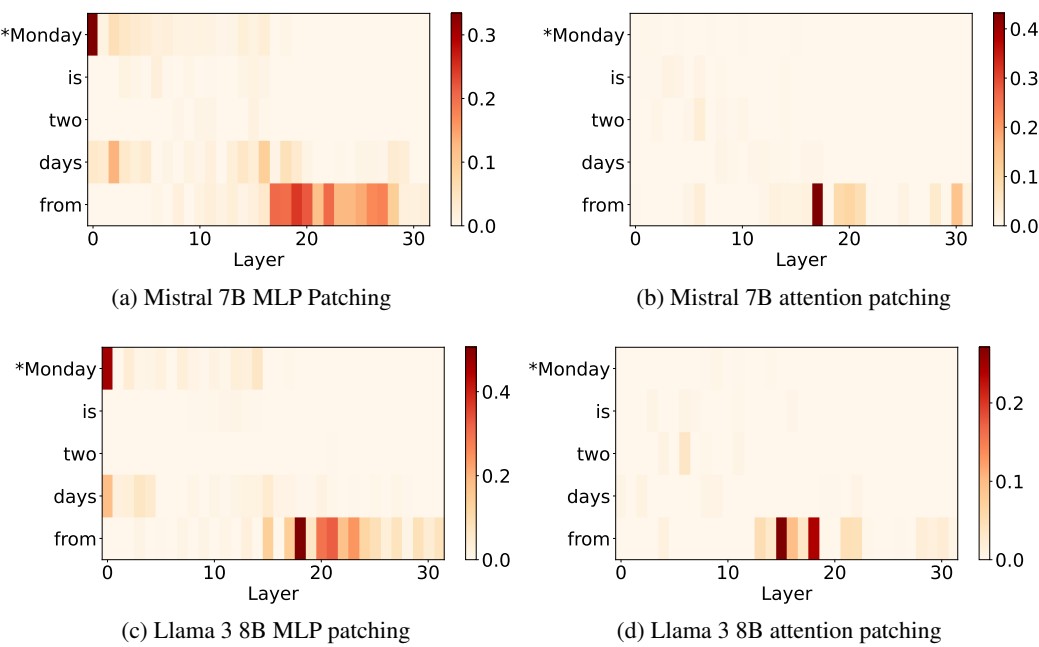

Figure 20: Attention and MLP patching results on `Weekdays`. Results are averaged over 20 different runs with fixed $\alpha$ and varying $\beta$ and 20 different runs with fixed $\beta$ and varying $\alpha$.

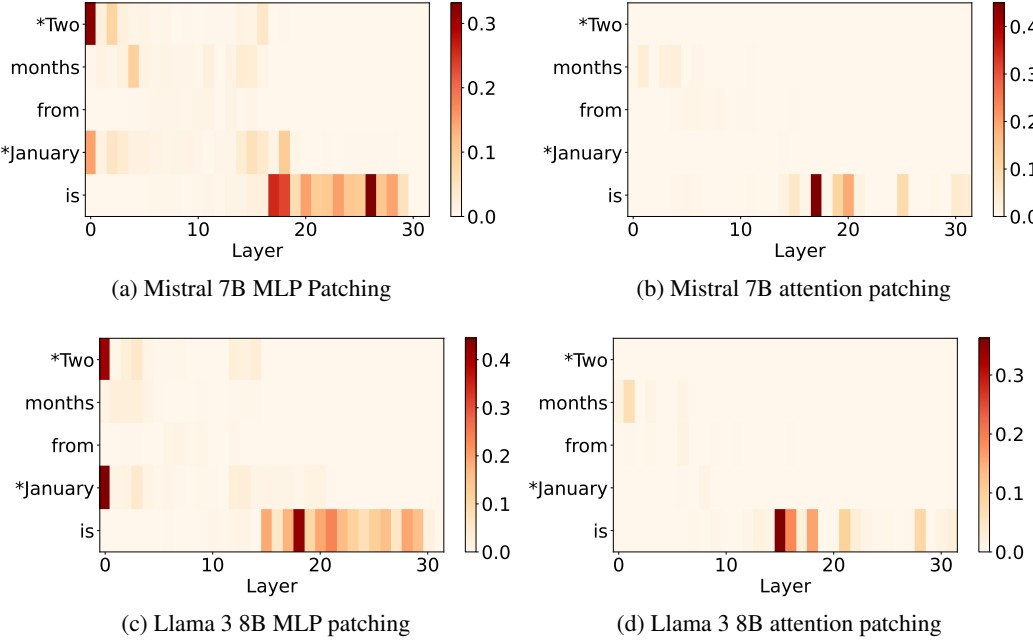

Figure 21: Attention and MLP patching results on `Months`. Results are averaged over 20 different runs with fixed $\alpha$ and varying $\beta$ and 20 different runs with fixed $\beta$ and varying $\alpha$.

### I.0.3 BASIC PATCHING

In Fig. 20 and Fig. 21, we report MLP and attention head patching results for `Weekdays` and `Months`. We experiment on 20 pairs of problems with the same $\alpha$ and different $\beta$ and 20 pairs of problems with the same $\beta$ and different $\alpha$, for a total of 40 pairs of problems. For each pair

Table 2: `Weekdays` finegrained results. Row ommited if both models get it correct.

| $\alpha$ | $\beta$ | Ground truth $\gamma$ | Mistral top $\gamma$ | Mistral correct? | Llama top $\gamma$ | Llama correct? |
|---|---|---|---|---|---|---|
| 1 | 1 | Wednesday | Wednesday | Yes | Thursday | No |
| 3 | 1 | Friday | Friday | Yes | Tuesday | No |
| 4 | 1 | Saturday | Saturday | Yes | Thursday | No |
| 3 | 2 | Saturday | Saturday | Yes | Tuesday | No |
| 4 | 2 | Sunday | Sunday | Yes | Wednesday | No |
| 5 | 2 | Monday | Monday | Yes | Tuesday | No |
| 2 | 3 | Saturday | Friday | No | Saturday | Yes |
| 3 | 3 | Sunday | Sunday | Yes | Tuesday | No |
| 4 | 3 | Monday | Monday | Yes | Tuesday | No |
| 0 | 4 | Friday | Thursday | No | Friday | Yes |
| 3 | 4 | Monday | Monday | Yes | Tuesday | No |
| 0 | 5 | Saturday | Friday | No | Saturday | Yes |
| 1 | 5 | Sunday | Saturday | No | Wednesday | No |
| 2 | 5 | Monday | Sunday | No | Monday | Yes |
| 4 | 5 | Wednesday | Tuesday | No | Tuesday | No |
| 6 | 5 | Friday | Thursday | No | Thursday | No |
| 1 | 6 | Monday | Sunday | No | Thursday | No |
| 2 | 6 | Tuesday | Monday | No | Tuesday | Yes |
| 3 | 6 | Wednesday | Tuesday | No | Tuesday | No |
| 4 | 6 | Thursday | Thursday | Yes | Tuesday | No |
| 5 | 6 | Friday | Friday | Yes | Thursday | No |
| 6 | 6 | Saturday | Thursday | No | Thursday | No |
| 0 | 7 | Monday | Sunday | No | Tuesday | No |
| 1 | 7 | Tuesday | Sunday | No | Tuesday | Yes |
| 2 | 7 | Wednesday | Sunday | No | Wednesday | Yes |
| 3 | 7 | Thursday | Sunday | No | Thursday | Yes |
| 4 | 7 | Friday | Thursday | No | Tuesday | No |
| 5 | 7 | Saturday | Friday | No | Saturday | Yes |
| 6 | 7 | Sunday | Friday | No | Thursday | No |

of problems, we patch the MLP/attention outputs from the "clean" to the "dirty" problem for each layer and token, and then complete the forward pass. Defining the logit difference as the logit of the clean $\gamma$ minus the logit of the dirty $\gamma$, we record what percent of the difference between the original logit difference of the dirty problem and the logit difference of the clean problem is recovered upon intervening, and average across these 40 percentages for each layer and token. This gives us a score we call the *Average Intervention Effect*.

For simplicity of presentation, we clip all of the (few) negative intervention averages to 0 (prior work (Zhang & Nanda, 2023) has also found negative-effect attention heads during patching experiments).

### I.0.4  CIRCLE CONTINUITY

Finally, in Fig. 22, we show another example of the continuity of the circular days of the week representation in Mistral 7B.

## J  PATCHING

In this section, we present results to support a claim that MLPs (and not attention blocks) are responsible for computing $\gamma$. In Fig. 25, we deconstruct states on top of the final token (before predicting $\gamma$) on Llama 3 8B `Months` (we show a similar plot for the states on the final token of Mistral 7B on `Weekdays` in the main text in Fig. 24. These plots show that the value of $\gamma$ is computed on the final token around layers 20 to 25. To show that this computation of occurs in the MLPs, we must show that no attention head is copying $\gamma$ from a prior token or directly computing $\gamma$.

Table 3: `Months` finegrained results. Row ommited if both models get it correct.

| $\alpha$ | $\beta$ | Ground truth $\gamma$ | Mistral top $\gamma$ | Mistral correct? | Llama top $\gamma$ | Llama correct? |
|---|---|---|---|---|---|---|
| 0 | 4 | May | April | No | May | Yes |
| 6 | 4 | November | October | No | November | Yes |
| 0 | 6 | July | June | No | July | Yes |
| 0 | 7 | August | July | No | August | Yes |
| 1 | 7 | September | October | No | September | Yes |
| 3 | 7 | November | October | No | November | Yes |
| 5 | 7 | January | December | No | January | Yes |
| 6 | 7 | February | January | No | February | Yes |
| 7 | 7 | March | February | No | March | Yes |
| 9 | 7 | May | April | No | May | Yes |
| 4 | 9 | February | February | Yes | January | No |
| 2 | 10 | January | December | No | January | Yes |
| 8 | 10 | July | June | No | July | Yes |
| 1 | 11 | January | December | No | January | Yes |
| 2 | 11 | February | December | No | February | Yes |
| 3 | 11 | March | February | No | March | Yes |
| 7 | 11 | July | June | No | July | Yes |
| 8 | 11 | August | July | No | August | Yes |
| 9 | 11 | September | August | No | September | Yes |
| 0 | 12 | January | December | No | January | Yes |

We first perform a patching experiment with the same setup Fig. 21 and Fig. 20 on individual attention heads on the final token. From the patching results we identify the top 10 attention heads by average intervention effect. For each attention head, we compute one EVR run with explanatory functions equal to one-hot functions of $\alpha$ and $\beta$ (resulting in 14 functions $\mathbf{g}_i$ for `Weekdays` and 24 for `Months`) and one with explanatory functions equal to one-hot functions of $\alpha$, $\beta$, and $\gamma$. We find that for all layers before 25, adding $\gamma$ to the explanatory functions adds almost no explanatory power. Since we established above that the model has already computed $\gamma$ at this point, we know that attention heads do not participate in computing $\gamma$.

To isolate the rough circuit for `Weekdays` and `Months`, we perform layer-wise activation patching on 40 random pairs of prompts. The results, displayed in Fig. 21 show

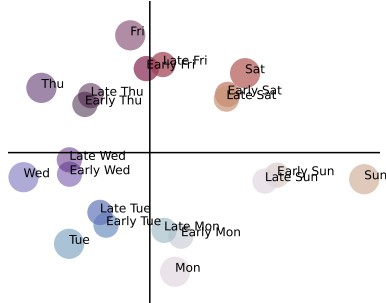

Figure 22: Layer 30 Mistral 7B activations for [very early/very late] on [Monday/Tuesday/.../Sunday], plotted projected into the PCA plane for [Monday/Tuesday/.../Sunday].

that the circuit to compute $\gamma$ consists of MLPs on top of the $\alpha$ and $\beta$ tokens, a copy to the token before $\gamma$, and further MLPs there (roughly similar to prior work studying arithmetic circuits (Stolfo et al., 2023)). Moreover, fine-grained patching in Appendix K shows that there are just a few responsible attention heads for the writes to the token before $\gamma$. However, patching alone cannot tell use *how* or *where* $\gamma$ is represented. For that, we need a new technique, which we expand on in the next section.

## K    EXPLANATION VIA REGRESSION (EVR)

So far, we have focused on examining and intervening on the representation for $\alpha$, which we present as a circle in the top PCA components on top of the $\alpha$ token. In this section, we examine how the generated output, $\gamma$, is represented.

First, to isolate the rough circuit for `Weekdays` and `Months`, we perform layer-wise activation patching on 40 random pairs of prompts. The results, displayed in Fig. 21 and Fig. 20, show that the circuit to compute $\gamma$ consists of MLPs on top of the $\alpha$ and $\beta$ tokens, a copy to the token before

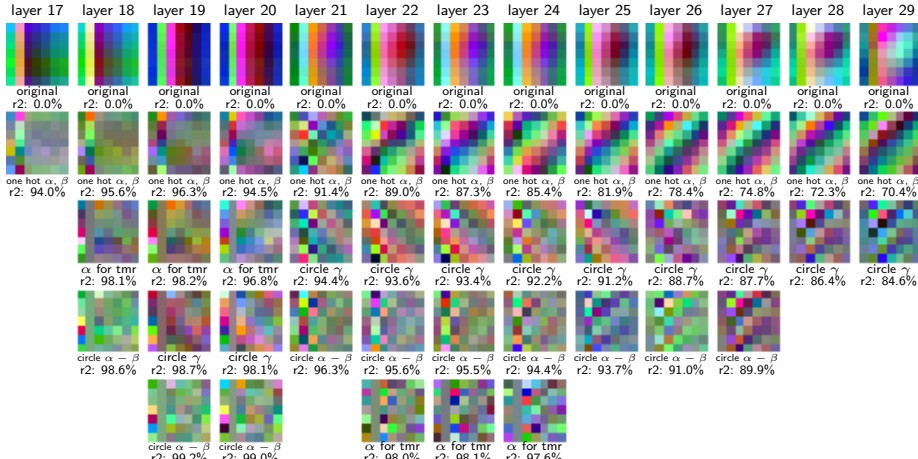

Figure 24: EVR residual RGB plots on Mistral hidden states on the `Weekdays` final token, layers 17 to 29. From top to bottom, we show each residual RGB plot after adding the function(s) $\mathbf{g}_i$ labelled just underneath, as well as the resulting $r^2$ value. We write "tmr" meaning "tomorrow" for $\beta = 1$. We also write "circle for $x$" meaning the inclusion of two functions $\mathbf{g}_i(x) = \{\cos, \sin\}(2\pi x/7)$.

$\gamma$, and further MLPs there (roughly similar to what Stolfo et al. (2023) find in prior work studying arithmetic circuits). Thus, we know *where* to look for a representation of $\gamma$: in the second half of the layers on the token before $\gamma$. However, patching alone cannot tell use *how* $\gamma$ is represented.

Unlike $\alpha$, $\gamma$ has no obvious circular (or linear) pattern in the top PCA components on these layers. To determine the representation for $\gamma$, we introduce a more powerful technique we call Explanation via Regression (EVR): given a set of token sequences with a corresponding set of hidden states $X_{i,l}$, we choose a set of interpretable explanation functions of the input tokens $\{\mathbf{g}_j(t)\}$. The $r^2$ value of a linear regression from $\{\mathbf{g}_j(t)\}$ to $X_{i,l}$ tells us how much of the variance in the activations the $\{\mathbf{g}_j(t)\}$ explain, and conversely the residuals show the exact components of the representation we have yet to explain.

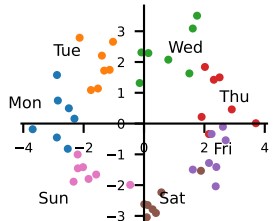

Figure 23: Top two PCA components of residual errors after EVR with one-hot in $\alpha$ and $\beta$. Mistral 7B `Weekdays`, layer 25, final token. Colored by $\gamma$.

### K.1 USING EVR TO UNCOVER A CIRCULAR REPRESENTATION FOR $\gamma$

We first use EVR to determine the representation for $\gamma$ by plotting the top two PCA components of the layer 25 Mistral 7B activations after subtracting the components that can be explained using a regression with one hot functions in $\alpha$ and $\beta$ (i.e. $\mathbf{g}_1 = [\alpha = 0], \mathbf{g}_2 = [\beta = 1], \mathbf{g}_3 = [\alpha = 1], \ldots$). The result, shown in Fig. 23, is an incredibly clear circle in $\gamma$, which suggests that the model's generated representation of $\gamma$ lies along a circle. A simple PCA projection was not enough to find this result because the representation for $\gamma$ has interference from $\alpha$ and $\beta$, which the EVR removes. This suggests that the models may be generating $\gamma$ by using a trigonometry based algorithm like the "clock" Nanda et al. (2023a) or "pizza" Zhong et al. (2024) algorithm in late MLP layers.

### K.2 MORE EXPERIMENTS WITH EVR

We now apply EVR to `Months` and `Weekdays` to break down $X_{i,l}$ completely into interpretable functions. We build a list of $\mathbf{g}_i$ *iteratively* and *greedily*. At each iteration, we perform a linear regression with the current list $\mathbf{g}_1 \ldots \mathbf{g}_k$, visualize and interpret the residual prediction errors, and build a new function $\mathbf{g}_{k+1}$ representing these errors to add to the list. Once most variance is explained, we can conclude that $\mathbf{g}_1, \ldots, \mathbf{g}_k$ constitutes the entirety of what is represented in the hidden states.

This information tells us what can and cannot be extracted via a linear probe, without having to train any probes. Furthermore, if we treat each $\mathbf{g}_i$ as a *feature* (see Definition 1), then the linear regression coefficients tell us which directions in $X_{i,l}$ these features are represented in, connecting back to Hypothesis 2.

Since $X_{i,l}$ consists of modular addition problems with two inputs $\alpha$ and $\beta$, we can visualize the errors as we iteratively construct $\mathbf{g}_1, \ldots, \mathbf{g}_k$ by making a heatmap with $\alpha$ and $\beta$ on the two axes, where the color shows what kind of error is made. More specifically, we take the top 3 PCA components of the error distribution and assign them to the colors red, green, and blue. We call the resulting heatmap a **residual RGB plot**. Errors that depend primarily on $\alpha$, $\beta$, or $\gamma$ show up as horizontal, vertical, or diagonal stripes on the residual RGB plot.

In Fig. 24, we perform EVR on the layer 17-29 hidden states of Mistral 7B on the `Weekdays` task; additional deconstructions are in Appendix K. We find that a circle in $\gamma$ develops and grows in explanatory power; we plot the layer 25 residuals after explaining with one hot functions in $\alpha$ and $\beta$ (i.e. $\mathbf{g}_1 = [\alpha = 0], \mathbf{g}_2 = [\beta = 1], \mathbf{g}_3 = [\alpha = 1], \ldots$) in Fig. 23 to show this incredibly clear circle in $\gamma$. This suggests that the models may be generating $\gamma$ by using a trigonometry based algorithm like the "clock" (Nanda et al., 2023a) or "pizza" (Zhong et al., 2024) algorithm in late MLP layers.

Table 4: Highest intervention effect attention heads from fine-grained attention head patching, as well as EVR results with one hot $\alpha, \beta$ and one hot $\alpha, \beta, \gamma$.

(a) Mistral 7B, `Weekdays`.

| L | H | Average Intervention Effect | EVR $R^2$ One Hot $\alpha, \beta$ | EVR $R^2$ One Hot $\alpha, \beta, \gamma$ |
|---|---|---|---|---|
| 28 | 18 | 0.22 | 0.39 | 0.73 |
| 18 | 30 | 0.17 | 0.95 | 0.96 |
| 15 | 13 | 0.17 | 0.94 | 0.95 |
| 22 | 15 | 0.11 | 0.77 | 0.82 |
| 16 | 21 | 0.09 | 0.92 | 0.93 |
| 28 | 16 | 0.08 | 0.42 | 0.69 |
| 15 | 14 | 0.06 | 0.98 | 0.99 |
| 30 | 24 | 0.05 | 0.43 | 0.79 |
| 21 | 26 | 0.04 | 0.53 | 0.63 |
| 14 | 2 | 0.04 | 0.93 | 0.95 |

(b) Llama 3 8B, `Weekdays`.

| L | H | Average Intervention Effect | EVR $R^2$ One Hot $\alpha, \beta$ | EVR $R^2$ One Hot $\alpha, \beta, \gamma$ |
|---|---|---|---|---|
| 17 | 0 | 0.18 | 0.98 | 0.99 |
| 17 | 1 | 0.08 | 0.98 | 0.98 |
| 19 | 10 | 0.08 | 0.95 | 0.96 |
| 30 | 17 | 0.07 | 0.85 | 0.90 |
| 17 | 3 | 0.07 | 0.93 | 0.95 |
| 17 | 27 | 0.06 | 1.00 | 1.00 |
| 31 | 22 | 0.05 | 0.37 | 0.78 |
| 21 | 9 | 0.04 | 0.73 | 0.78 |
| 20 | 28 | 0.04 | 1.00 | 1.00 |
| 30 | 16 | 0.04 | 0.73 | 0.85 |

(c) Mistral 7B, `Months`.

| L | H | Average Intervention Effect | EVR $R^2$ One Hot $\alpha, \beta$ | EVR $R^2$ One Hot $\alpha, \beta, \gamma$ |
|---|---|---|---|---|
| 20 | 28 | 0.15 | 0.76 | 0.76 |
| 17 | 0 | 0.10 | 0.77 | 0.77 |
| 25 | 14 | 0.08 | 0.19 | 0.61 |
| 17 | 1 | 0.07 | 0.80 | 0.82 |
| 17 | 3 | 0.06 | 0.71 | 0.71 |
| 31 | 22 | 0.06 | 0.12 | 0.67 |
| 17 | 27 | 0.05 | 0.58 | 0.58 |
| 19 | 4 | 0.05 | 0.40 | 0.66 |
| 19 | 10 | 0.04 | 0.62 | 0.62 |
| 30 | 26 | 0.04 | 0.51 | 0.62 |

(d) Llama 3 8B, `Months`.

| L | H | Average Intervention Effect | EVR $R^2$ One Hot $\alpha, \beta$ | EVR $R^2$ One Hot $\alpha, \beta, \gamma$ |
|---|---|---|---|---|
| 15 | 13 | 0.26 | 0.62 | 0.62 |
| 16 | 21 | 0.17 | 0.76 | 0.76 |
| 18 | 30 | 0.13 | 0.77 | 0.77 |
| 28 | 18 | 0.11 | 0.13 | 0.52 |
| 28 | 16 | 0.07 | 0.13 | 0.52 |
| 21 | 25 | 0.05 | 0.65 | 0.70 |
| 15 | 14 | 0.03 | 0.72 | 0.72 |
| 17 | 26 | 0.02 | 0.77 | 0.77 |
| 31 | 1 | 0.02 | 0.11 | 0.57 |
| 21 | 24 | 0.02 | 0.30 | 0.45 |

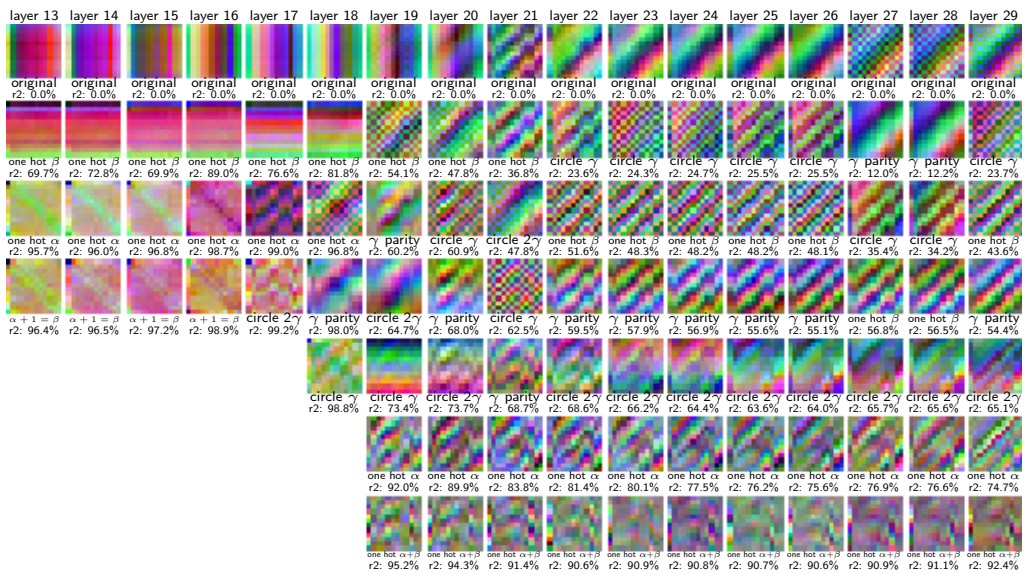

Figure 25: Iterative deconstruction of hidden state representations on the final token on Llama 3 8B, `Months`.

