# OpenReview forum: "Not All Language Model Features Are One-Dimensionally Linear"
_ICLR.cc/2025/Conference — ICLR 2025 Poster_

### Official Review · Reviewer_hTfC · 2024-11-03

**Soundness:** 3
**Presentation:** 3
**Contribution:** 3
**Rating:** 8
**Confidence:** 3

**Summary:**

The authors show convincingly that large language models have circular representations of periodic variables, and provide a means of discovering and probing these representations using sparse autoencoders and causal interventions.

**Strengths:**

The results are straightforward, and backed up by causal experiments. Clearly connects to existing methods in the interpretability field, which could make it simple for others to replicate.

**Weaknesses:**

The clustering method was a little hard to follow for me at first, maybe pseudocode would be helpful (since it seems like it wouldn't be many lines).

**Questions:**

Is it mostly just circles? Given that they are still a one-dimensional variable, I'm wondering whether there are representations of variables with (irreducible) intrinsic dimension higher than one (like a spherical variable), which would seem to make better use of the scope of your Hypothesis 2. Otherwise, maybe the linear representation hypothesis has been violated in name but not in spirit (i.e. representation is still factorized into one-dimensional variables)?

---

> ### Author Response · Authors · 2024-11-19
> **Response**
>
> Thank you for your thoughtful comments, we are glad you found our paper interesting!
>
> All changes we discuss below are in the new submitted revision of our paper. Where possible, we have highlighted changes in the revised document that are specific to your comments in *teal*.
>
> > The clustering method was a little hard to follow for me at first, maybe pseudocode would be helpful (since it seems like it wouldn't be many lines).
>
> Thank you for this excellent suggestion! We have added an algorithm block to Appendix F, as well as a link to this pseudocode from Section 4. Please let us know if anything is still unclear!
>
> > Is it mostly just circles? Given that they are still a one-dimensional variable, I'm wondering whether there are representations of variables with (irreducible) intrinsic dimension higher than one (like a spherical variable), which would seem to make better use of the scope of your Hypothesis 2. Otherwise, maybe the linear representation hypothesis has been violated in name but not in spirit (i.e. representation is still factorized into one-dimensional variables)?
>
> Our method is limited to finding two-dimensional representations, so we do not identify spheres or other 3D objects. That being said, we have added the top ranked clusters to Appendix G (Figure 16) that our method discovered, and some of them look like they might have 2D intrinsic structure, although it is hard to tell. Additionally, recent work has found a globe representation in an LLM[1], so we believe these higher dimensional features certainly exist. We have added a comment about this in our discussion, thank you for the suggestion!
>
> [1] https://www.lesswrong.com/posts/guNzr32FC6DYCaSgJ/there-is-a-globe-in-your-llm

---

### Official Review · Reviewer_9dQp · 2024-11-04

**Soundness:** 3
**Presentation:** 3
**Contribution:** 3
**Rating:** 6
**Confidence:** 4

**Summary:**

The authors define multi-dimensional features and identify these representations empirically using sparse autoencoders. They also show that some LLMs use circular representations for days of the week and months of the year to perform modular addition.

**Strengths:**

The extension of the linear representation hypothesis from one-dimensional to multi-dimensional one provides good insights for researchers in the mechanistic interpretability field. Concrete and extensive empirical results show that multi-dimensional features exist in LLMs. In particular, the circular representations of days of the week are interesting.

**Weaknesses:**

Some details in formalization and experiments are not clear. Please see the questions below.

**Questions:**

1. As you mentioned, the title is confusing. How about 'Not All Language Model Features are One-dimensionally Linear'?
2. Why do you use the term 'point cloud' in Definition 1? Does it mean that the range of the map $f$ is a discrete set?
3. What is the source of randomness for $p(a,b)$ in Definition 2? Does it come from the distribution of input texts (in training set for LLMs)? Also, why do you state that statistical independence and separability are equivalent?
4. In Figure 2, the histogram appears to show a mixture. Why did you say that the $\epsilon$-mixture index of 0.4750 is low? What is the baseline? In Figure 10, you state that an index of 0.3621 indicates the feature is likely a mixture. This seems contradictory.
5. In line 269, why do you state that 'These dictionary elements will then have a high cosine similarity'? I believe they could have low cosine similarity even if many elements span the feature.
6. In Figure 1, how were the points labeled? What does the label 'Monday' mean? Does it indicate that 'Monday' would be the next token in the corresponding texts?
7. In Section 5, you use specific texts and train new probes. Why didn't you use the circular representations previously identified by SAE clustering?

---

> ### Author Response · Authors · 2024-11-19
> **Response**
>
> Thank you for your helpful comments and feedback!
>
> All changes we discuss below are in the new submitted revision of our paper. Where possible, we have highlighted changes in the revised document that are specific to your comments in *blue*.
>
> **Responses to questions:**
>
> > As you mentioned, the title is confusing. How about 'Not All Language Model Features are One-dimensionally Linear'?
>
> This is a great suggestion, thank you! We have made this change.
>
> > Why do you use the term 'point cloud' in Definition 1? Does it mean that the range of the map f is a discrete set?
>
> Thank you for pointing this out, we agree this is confusing and have removed it, a feature is now simply defined as a function from the token input space to the model’s hidden space.
>
> > What is the source of randomness for $p(a,b)$ in Definition 2? Does it come from the distribution of input texts (in training set for LLMs)? Also, why do you state that statistical independence and separability are equivalent?
>
> Regarding randomness, yes, it comes from the input text distribution $\mathcal{T}$. We have made this more clear by describing $\mathcal{T}$ explicitly at the beginning of Section 3. Regarding the equivalence of statistical independence and separability, we are just defining separability to mean that the distribution can be decomposed into (lower dimensional) independent distributions, which we believe is a common choice for the meaning of “separable distribution”. Thank you for pointing both of these out, let us know if anything is still unclear!
>
> > In Figure 2, the histogram appears to show a mixture. Why did you say that the $\epsilon$-mixture index of 0.4750 is low? What is the baseline? In Figure 10, you state that an index of 0.3621 indicates the feature is likely a mixture. This seems contradictory.
>
> In Figure 2, we are comparing to other GPT-2 clusters that we have discovered, where an $\epsilon$-mixture index of 0.4750 is indeed low (see Figure 3). In the appendix we are working with synthetic distributions that are cleaner, so the mixture measures are lower (e.g. they do not have a circle of noisy points very near the origin). We agree this is confusing and have tweaked the example to be more of a mixture, so now the measured $\epsilon-mixture$ index in Figure 10a is $~0.64$. Thank you for pointing this out!
>
> > In line 269, why do you state that 'These dictionary elements will then have a high cosine similarity'? I believe they could have low cosine similarity even if many elements span the feature.
>
> It is true that not all pairwise vectors in the space will have high cosine similarity, but we believe that adjacent pairs radially will likely have high cosine similarity since it is not possible to “pack in” many features in 2D all of which have pairwise low cosine similarity. Could you be more specific about the setup you are worried about?
>
> > In Figure 1, how were the points labeled? What does the label 'Monday' mean? Does it indicate that 'Monday' would be the next token in the corresponding texts?
>
> Our apologies for not being more clear, the label corresponds to the current token, we have added this clarification to the Figure 1 caption.
>
> > In Section 5, you use specific texts and train new probes. Why didn't you use the circular representations previously identified by SAE clustering?
>
> This is a great point, and is a result of the order in which we originally ran experiments. Empirically, the two planes (one from the probe, the other from the PCA of the SAE cluster) are highly similar. The principal angles between the planes are 45 and 50 degrees, which means that projections of basis vectors from one plane into the other maintain > cos(45) = 66% of their norm. Thus we expect the results of this experiment to be similar, although we have not run this experiment.
>
> If our comments resolve your concerns we would appreciate if you would consider raising your score!

---

> > ### Comment · Reviewer_9dQp · 2024-11-25
> >
> > Thank you for your helpful response and revision!
> >
> > I’m still concerned that the randomness is not thoroughly defined. As you mentioned, the input token distribution induces the feature probability distribution $p(a,b)$ for one feature. But is this a probability density function or a cumulative distribution function? Also, is the integral of $p(a,b)$ equal to 1? As you mentioned, only a few parts of the text activate the feature. Is $p(a,b)$ a conditional probability given those specific parts of the text?
> >
> > Regarding the histogram in Figure 2, it still looks like a mixture, and the value near 0.5 appears to remain within the GPT-2 cluster. How did you determine the threshold for reducibility?
> >
> > As for Section 5, I don’t think the results are similar. The remaining dimensions could influence the intervention. Did you run the experiments with the circular representations previously identified through SAE clustering? Is there the result in the Appendix?

---

> > > ### Author Response · Authors · 2024-11-26
> > >
> > > Thank you for these suggestions and questions, we have uploaded a new revision of the paper and below respond to your comments. Let us know if anything is still unclear!
> > >
> > > > I’m still concerned that the randomness is not thoroughly defined. As you mentioned, the input token distribution induces the feature probability distribution $p(a, b)$ for one feature. But is this a probability density function or a cumulative distribution function? Also, is the integral of $p(a, b)$ equal to $1$? As you mentioned, only a few parts of the text activate the feature. Is $p(a, b)$ a conditional probability given those specific parts of the text?
> > >
> > > Yes, $p(a, b)$ is a conditional probability density function that has an integral equal to 1. In other words, it is the probability that the feature $f$ equals a certain value across the input distribution, conditioned on the feature activating. We have added a line clarifying this after Definition 2 in the revision, let us know if this helps!
> > >
> > > > Regarding the histogram in Figure 2, it still looks like a mixture, and the value near 0.5 appears to remain within the GPT-2 cluster. How did you determine the threshold for reducibility?
> > >
> > > Since we have two measures of reducibility–mixture and separability–we do not fix a threshold for either, but instead examine their product to determine a ranking of GPT-2 clusters by their reducibility. We describe this metric at the end of Section 4; it is defined as
> > > $$reducibilityScore = \sum_{i} (1 - mixture_i) * (separability_i)$$
> > > where $i$ indices into pairs of PCA dimensions. The clusters shown in Figure 1 rank $9$, $28$, and $15$ out of all $1000$ clusters by this combined metric, respectively. Specifically, the days of the week cluster in Figure 2 ranks $9$. We have added labels to the plot in Figure 3 to help clarify this.
> > >
> > > > As for Section 5, I don’t think the results are similar. The remaining dimensions could influence the intervention. Did you run the experiments with the circular representations previously identified through SAE clustering? Is there the result in the Appendix?
> > >
> > > Thank you for pushing us to add this experiment, we have now added it in the new revision of the paper, and we believe it makes our story significantly stronger, as now each section connects directly to the next. Specifically, in the new revision of the paper, we have compared using the learned probe of the day of the week versus using the plane identified by PCA dims 2 and 3 from the Mistral days of the week cluster.
> > >
> > > We run only on layer 8 and neighboring layers, since layer 8 is the only at which we have Mistral SAE clustering results and because the SAE-identified subspace remains valid only on nearby layers due to representation drift, see e.g. [1]. We find that using the exact circular representations identified through SAE clustering only slightly decreases intervention performance as compared to training a probe (from -2.58 to -2.01 average logit difference on layer 8). Even more interestingly, the layer 8 SAE representation space is much more robust to layer changes than the layer 8 learned probe space; for example, using the layer 8 learned probe subspace on layer 6 results in an average logit difference of 0.029, whereas using the layer 8 SAE probe space on layer 6 results in a much better average logit difference of -2.32. We believe this is intriguing evidence that the SAE is perhaps finding more “true” features; thank you again for suggesting this experiment!
> > >
> > > We have added this result and accompanying discussion to Appendix I.0.2 (figure 19 and figure 20) and have referenced it in Section 5.1.
> > >
> > > [1] Eliciting Latent Predictions from Transformers with the Tuned Lens, Belrose et. al.

---

> > > > ### Comment · Reviewer_9dQp · 2024-11-30
> > > >
> > > > Thank you for your response!
> > > >
> > > > I think it would improve the flow of the main text to include the intervention experiments with the SAE clustering in the main text and move the original experiments to the appendix.

---

> > > > > ### Author Response · Authors · 2024-12-02
> > > > >
> > > > > Thank you for your suggestion! Since we only have SAE clustering results for layer 8 of Mistral, we cannot apply the SAE intervention across all layers and on Llama, which we believe is an important part of the paper that deserves to be in the main body. We also feel that the technique of circular probing is important to highlight. However, we agree that these are important experiments that make the paper more cohesive, so we propose to add the new experiments using the SAE circle plane to the main body in section 5 in addition to the learned probe experiments, add an explanation at the start of section 5 that we will do two types of interventions (learned probes and SAE based probes), and put the experiments from section 5.2 in the appendix if we cannot make space. We cannot submit a new revision with these changes as the deadline for new revisions has passed, so we can only describe this change. Does this sound reasonable? Thank you again!

---

> > > > > > ### Comment · Reviewer_9dQp · 2024-12-02
> > > > > >
> > > > > > Thank you for your response! That sounds reasonable.
> > > > > >
> > > > > > I’m still concerned about the criterion for irreducibility. Reducibility is defined as a feature that “satisfies at least one of the conditions,” which means irreducibility requires the feature to fail both conditions.
> > > > > >
> > > > > > Therefore, the product of the scores would not be the correct criterion for irreducibility. Instead, a feature is considered irreducible only if each condition is independently unmet based on its respective criterion. Accordingly, the irreducible points should lie along the regions where the right and top boundaries of the cluster intersect in Figure 3.

---

> > > > > > > ### Author Response · Authors · 2024-12-02
> > > > > > >
> > > > > > > Great point! We used the product formulation because it was simple to explain and was one reasonable way to trade off between the two metrics to establish a ranking of clusters by irreducibility.  We agree, however, that a stricter metric may match the definitions better. Thus, we reran with $$score = \sum_i min(rank(mixture_i), rank(1 - seperability_i))$$ Here, $i$ still indexes into pairs of PCA dims, and $rank$ means that we are taking the ranking of the cluster among values of that metric for all clusters for the given pair of PCA dimensions. Weekdays, months, and years were then ranked [8, 105, 12] respectively. Thus, ranking in this manner still seems to work well!
> > > > > > >
> > > > > > > Since the exact way we balance against the two metrics is a heuristic and not specified in the definition, we plan to keep the current metric in the paper the same, both because it is simpler to explain in the main body and because Figure 17 would require a significant refactor if we changed the ranking metric. We will add text to the main body pointing to an appendix where we discuss trading off these metrics, describe this alternative ranking scheme, and describe the resulting ranks of weekdays, months, and years.
> > > > > > >
> > > > > > > Thank you again!

---

### Official Review · Reviewer_d8Uh · 2024-11-04

**Soundness:** 3
**Presentation:** 3
**Contribution:** 3
**Rating:** 8
**Confidence:** 3

**Summary:**

This work's main focus is to set a theoretical framework for irreducible multidimensional features and to show their concrete existence in real-life LLMs. In particular, the authors:
- update the superposition definition to account for irreducible multi-dimensional features.
- propose an algorithm that---using an SAE-learned dictionary---identifies possible multi-dimensional features.
- show evidence of features expressing circular behavior for the representation (and manipulation) of periodic data, such as months, weeks, and days. The models whose features are shown to exist are non-toy models, such as Llama 3 8B and Mistral 7B, further increasing their argument's validity.

**Strengths:**

1) The updated proposed definition for the superposition hypothesis is sound and may lead to the discovery of other interesting structures such as the ones presented in the paper. Even if this direction does not lead to further scientific discoveries, the discovered representations themselves are an interesting finding.

2) In the paper, features are extracted from state-of-the-art LLMs, showcasing the existence of actual multi-dimensional features of circular nature **"in the wild"**.

**Weaknesses:**

1) If I understand correctly, the proposed algorithm can extract interpretable features, however, I imagine there is a good amount of them that is not easily--if at all--interpretable. Adding some of your insight on how many potentially interesting multi-dimensional features are among the ones extracted from the algorithm could improve the article.

2) If I am not mistaken, there is not an ablation on how much the threshold parameter T affects the extracted clusters: Having an understanding of when circular features start to emerge could be an interesting experiment to add, and may potentially lead to some useful insights.

3) Figures 2 and 3 can be improved in their overall presentation:
- Figure 2: I would change the color for the PCAs scatter plot from black to something with less contrast (e.g. the green used in the angle subfigure of  Figure 2)
- Figure 3:  This is a very minor complaint, but I would change the scatter icon for the "other cluster" points to something different than the one used to indicate the clusters shown in Figure 1, for example, something like an **x** would probably look a bit better.

4) Another minor issue is that the captions for Figures 5 and 6 are a bit lacking, adding some more information to help understand their respective feature would probably be helpful.

**Questions:**

1) Do you have any qualitative insights on how big the threshold parameter T should be in order to find multidimensional features?

2) Are there any extracted multi-dimensional features that show some interesting behavior, like the circular one shown in the paper? If not, do you think is it due to a structure difficult to interpret, or simply because probably most of the features are still one-dimensional?

---

> ### Author Response · Authors · 2024-11-19
> **Response**
>
> Thank you for your thoughtful comments, we are glad you found our paper interesting!
>
> All changes we discuss below are in the new submitted revision of our paper. Where possible, we have highlighted changes in the revised document that are specific to your comments in *orange*.
>
> **Responses to weaknesses:**
>
> > If I understand correctly, the proposed algorithm can extract interpretable features, however, I imagine there is a good amount of them that is not easily--if at all--interpretable. Adding some of your insight on how many potentially interesting multi-dimensional features are among the ones extracted from the algorithm could improve the article.
>
> We agree this is an interesting potential insight! We have added some of the other features discovered in Figure 17 in Appendix G. Most of them seem interpretable in terms of the tokens that they are composed of (common words and single letters), but their structure is not entirely interpretable. We believe this is important future work!
>
> > If I am not mistaken, there is not an ablation on how much the threshold parameter T affects the extracted clusters: Having an understanding of when circular features start to emerge could be an interesting experiment to add, and may potentially lead to some useful insights.
>
> Thank you for this excellent suggestion! We have added an experiment in Appendix F2 (Figure 16) that varies the threshold (and the k) for the Mistral clustering procedure and sees when the days of the week cluster is identifiable. We find that small T causes all features to group in one cluster, so the days of the week cluster is not found; medium T causes the days of the week cluster to become identifiable; and large T causes all features to be divided into their own clusters.
>
> > Figure 2: I would change the color for the PCAs scatter plot from black to something with less contrast (e.g. the green used in the angle subfigure of Figure 2)
>
> Thank you for this suggestion! We’ve made the scatter points grey with alpha=0.3 with a black edgecolor. This reduces the contrast and makes the relative density of the points easier to see.
>
> > Figure 3: This is a very minor complaint, but I would change the scatter icon for the "other cluster" points to something different than the one used to indicate the clusters shown in Figure 1, for example, something like an x would probably look a bit better.
>
> This for this suggestion too! We’ve marked them with a slightly larger x, and also made the orange color darker so they are easier to see.
>
> > Another minor issue is that the captions for Figures 5 and 6 are a bit lacking, adding some more information to help understand their respective feature would probably be helpful.
>
> We have added more information to these captions, thank you for this suggestion!
>
> **Responses to questions:**
>
> > Do you have any qualitative insights on how big the threshold parameter T should be in order to find multidimensional features?
>
> Now we do, see our response above, thank you! Empirically, T should be set such that it is right on the boundary between everything being one large cluster and everything being in its own cluster.
>
> > Are there any extracted multi-dimensional features that show some interesting behavior, like the circular one shown in the paper? If not, do you think is it due to a structure difficult to interpret, or simply because probably most of the features are still one-dimensional?
>
> This is a good suggestion, thank you! We did not do interventions on other multi-d features because their structure seems harder to interpret (see Figure 17 for more of these examples), but it is possible that further study of e.g. belief states (see [1]) will help us understand these structures.
>
> [1] https://www.lesswrong.com/posts/gTZ2SxesbHckJ3CkF/transformers-represent-belief-state-geometry-in-their

---

### Official Review · Reviewer_x6ai · 2024-11-10

**Soundness:** 3
**Presentation:** 2
**Contribution:** 3
**Rating:** 6
**Confidence:** 4

**Summary:**

The paper calls into question the Linear Representation Hypothesis (LRH), in particular the part that claims the representations of features in LLMs lie along one-dimensional lines.
A clear visual example is provided in Figure 1, where a circular feature for days of the week is shown in a 2-dimensional PCA space.
The paper then formally defines irreducible, multi-dimensional features and proposes an updated version of Elhage et al.'s Linear Superposition Hypothesis to incorporate multi-dimensional features.
The specific definition of irreducibility is further used to derive a clustering-based method to automatically find irreducible features among the dictionary elements learned by sparse autoencoders (SAEs) trained on LLMs.
Finally, the paper illustrates how steering the circular feature of day-of-week can alter the model's output in synthetic tasks.

**Strengths:**

- The paper tackles a timely and important question in mechanistic interpretability by formalizing (inherently) multi-dimensional features in LLMs.
- The visual examples of circular representations are clear and compelling.
- I think the high-level idea of formalizing multi-dimensional, irreducible features is sensible and useful.
- Overall, the experiments are comprehensive with a variety of interesting results.

**Weaknesses:**

- I think there are some conceptual questions about the definitions of features and irreducibility.
    - Conceptually, I think the definition of irreducibility is somewhat incomplete. First, there can be concepts that are correlated but can still be disentangled for separate interventions, so equating separability with independence is can be limiting. Second, I can't tell for sure if the definition of reducibility is exhaustive. (Can there be a third category?)
    - I have asked about some of the more specific unclear points in the questions below.
- While the paper's main motivation is to call the LRH into question, the proposed alternative hypothesis appears to be a stricter version of the LRH (p. 5, lines 239--240), rather than a contradiction to it. This makes the overall message a bit confusing. (I'd be happy to be corrected if this is a misunderstanding.)
- The proposed "tests" of irreducibility are not completely formal, in the sense that they are not statistical tests of significance but rather (possibly arbitrary) thresholding of a pair of metrics. This could mean that whether or not a feature is irreducible is ultimately a judgment call by the user.
    - To be clear, I'm not saying this is something that can be readily addressed; it's a limitation that should be mentioned somewhere.
- For the intervention tests, I'm not fully sure if the results demonstrate "causal implication" of the circular feature. The results are interesting, but I'm not sure if they are conclusive. See second-to-last question below.
- Despite its interesting results, the paper's exposition is not very clear.
    - I find that Section 3 is not easy to follow. The only visual aid is Figure 2, but it is not referenced or intuitive.
        - E.g., why exactly are we looking at PCA 2/3, even for Fig. 1? What is the histogram in the middle, and what is the angle on the right? What are "typical clusters" and what are their mixture/separability indices?
        - I believe some of these are answered in the referenced Appendix. While I am okay with most computational details being deferred to the Appendix, at least the plots should be self-contained. Also, some of the key computational details should be noted (that separability index is computed in 2-d and that mixture index is computed approximately via SGD). Otherwise, the exposition is quite hard to follow.
    - While some of the plots are great, the experiments are quite dense and unorganized. The exposition would often refer to a figure in the Appendix, while some figures (2 and 4) in the main text are not referenced. This makes it difficult to read the paper linearly.
        - I would recommend starting each subsection with a clear motivation, the key result(s), and then reference supplementary results in the Appendix.
        - Perhaps a few more plots can be moved to the Appendix in favor of more explanation/motivation/examples.

**Questions:**

- Figure 1: is there any intuition for why specifically we find these features in the second and third PCA components?
- Definition 1: What exactly is a $d_f$-dimensional point cloud in $\mathbb{R}^{d_f}$? Is there a mathematical definition? (Is $d_f$ redundant?) If I have two points in $\mathbb{R}^3$, what is the "dimension" of the point cloud they form?
- Also, to be precise, should I understand the "probability distribution over input tokens" as that of the natural language? I also assume that this probability is restricted to $n$-length token sequences---is that correct?
  - Later, in the definition of the two indices, the distribution comes up anyway as $\mathbb{P}$. It may be helpful to formally define this notation/distribution up front.
- Definition 2: in the mixture definition, what is the non-bold $b$? Also, does "lower dimensional" simply mean $k < d_f/2$ or does it mean low-dimensional, say, $k \ll d_f$?
- Is there intuition as to why it suffices to consider separable and disjoint features as the two types of reducible features? Could there be other types of reducible features?
- Can you efficiently compute the separability index for features that are beyond two-dimensional?
- Hypothesis 2: while this is a stylistic point, I think the use of a possessive in "Our Superposition Hypothesis" is unnecessary and un-scientific. It could have any other objective name, say, the "Multi-Dimensional Superposition Hypothesis". See, e.g., [Knuth et al.](https://jmlr.csail.mit.edu/reviewing-papers/knuth_mathematical_writing.pdf), page 2 item 6.
- p. 5, lines 260--270: doesn't this claim depend heavily on (1) the two-dimensional feature being active in the LLM's representation space and (2) the SAE is large enough to capture this feature? I'm not even sure if this claim is necessary, as opposed to saying "some of SAE clusters represent irreducible features".
- Figure 3: what are some of the other features that are shown  here to be more reducible than the features for Fig. 1?
- Are there any other features that we may intuitively view as circular but not found by the SAE? Say, are color words (red-orange-yellow-green-blue-purple) represented in a circular fashion?
- p. 8, "In practice, ...": this is somewhat confusing/concerning to me. If there are alternative pathways that may affect both the circular feature and the task output, then wouldn't they "wash away" the causal implication of the feature? In other words, in what sense exactly is this a "causal" implication rather than correlational?
- Broad question: are there examples of multi-dimensional features that aren't circular?

---

> ### Author Response · Authors · 2024-11-19
> **Response Part 1**
>
> Thank you for your thoughtful and helpful comments!
>
> All changes we discuss below are in the new submitted revision of our paper. Where possible, we have highlighted changes in the revised document that are specific to your comments in *red*.
>
> **Responses to weaknesses:**
>
> >First, there can be concepts that are correlated but can still be disentangled for separate interventions, so equating separability with independence can be limiting.
>
> Thank you for pointing this out! We agree that focusing on geometric and statistical based definitions might not tell the full story, but these definitions are helpful in that they can be applied to real features across a large corpus of text, as we do in the paper (although see Appendix C where we propose an intervention-based definition). We have added this point to our limitations discussion section in the paper.
>
> >Second, I can't tell for sure if the definition of reducibility is exhaustive. (Can there be a third category?)
>
> Our current definitions prevent multiple one-dimensional features that are either unrelated or anti-correlated from being treated as a multi-d feature (via separability index and mixture index, respectively). These cases seem like a natural breakdown of the types of statistical reducibility; if two features are neither of these things, then their co-firing pattern may have interesting structure.
>
> >While the paper's main motivation is to call the LRH into question, the proposed alternative hypothesis appears to be a stricter version of the LRH (p. 5, lines 239--240), rather than a contradiction to it. This makes the overall message a bit confusing. (I'd be happy to be corrected if this is a misunderstanding.)
>
> This is an excellent point! As suggested by Reviewer 9dQp, we have renamed the paper “Not All Language Model Features are One-Dimensionally Linear”, which we hope makes it more clear we are not claiming the entire LRH is false. We state that the multi-d hypothesis is a stricter version of the LRH to describe how the definitions are related, but technically any possible distribution of vectors could be seen as a linear combination of one dimensional vectors (e.g. the original neuron basis), it just might not be that sparse or mono-semantic, so it is hard to “contradict” the LRH. The multi-d hypothesis might be a more “true” breakdown of the space, in that the features are monosemantic and overall sparsity is lower.
>
> >The proposed "tests" of irreducibility are not completely formal, in the sense that they are not statistical tests of significance but rather (possibly arbitrary) thresholding of a pair of metrics. This could mean that whether or not a feature is irreducible is ultimately a judgment call by the user.
>
> We agree and this is an important point, thank you! We have added a discussion of this to the limitations section in our discussion.
>
> >I find that Section 3 is not easy to follow. The only visual aid is Figure 2, but it is not referenced or intuitive. E.g., why exactly are we looking at PCA 2/3, even for Fig. 1? What is the histogram in the middle, and what is the angle on the right? What are "typical clusters" and what are their mixture/separability indices?
>
> Thank you for pointing this out! We have added further explanation to Figure 2 describing each part of the figure and further explanation at the end of Section 3.1 that describes the implementations of the tests. Please let us know if this is still unclear! Regarding looking at PCA 2 and 3 instead of 1 and 2, we now describe this in section 4, see our response to Q1 below for more details.
>
> >While some of the plots are great, the experiments are quite dense and unorganized. The exposition would often refer to a figure in the Appendix, while some figures (2 and 4) in the main text are not referenced. This makes it difficult to read the paper linearly.
>
> We have now added references to figures 2 and 4, and have also added motivations to sections and subsections that were missing them. Thank you for suggesting this improvement!

---

> ### Author Response · Authors · 2024-11-19
> **Response Part 2**
>
> **Responses to questions, part 1:**
>
> >Figure 1: is there any intuition for why specifically we find these features in the second and third PCA components?
>
> Yes, the first dimension is typically an “intensity” direction that manifests as the radius in the circular plots. Thus the actual multi-dimensional structure might best be thought of as a cone.  We have added a discussion of this in section 4, thank you for inspiring this clarification!
>
> > Definition 1: What exactly is a df-dimensional point cloud in Rdf? Is there a mathematical definition? (Is df redundant?) If I have two points in R3, what is the "dimension" of the point cloud they form?
>
> We intended for $d_f$-dimensional point cloud to refer to a distribution that is “full rank,” i.e. one that does not lie on a lower dimensional subspace. We agree that this is not necessary, however, and so we have omitted it in the revision.
>
> > Also, to be precise, should I understand the "probability distribution over input tokens" as that of the natural language? I also assume that this probability is restricted to n-length token sequences---is that correct? Later, in the definition of the two indices, the distribution comes up anyway as P. It may be helpful to formally define this notation/distribution up front.
>
> Thank you for asking us to clarify this! Yes, this is correct, we have added a term $\mathcal{T}$ that represents the input text distribution and define this explicitly up front at the beginning of section 3. Note that $\mathbb{P}$ defined in equation 3 is meant to denote the “probability” operator over $\mathcal{T}$; we have made this more explicit in the current revision.
>
> > Definition 2: in the mixture definition, what is the non-bold b? Also, does "lower dimensional" simply mean $k<d_f/2$ or does it mean low-dimensional, say, $k \ll d_f$?
>
> The non-bold b is a typo and should be bold, thank you for catching that! Lower dimensional just means $k < d_f$. For example, consider a three dimensional feature that could be decomposed as mostly lying on one or more underlying planes; then those planes should be the features.
>
> > Can you efficiently compute the separability index for features that are beyond two-dimensional?
>
> In theory one could compute a separability index measure for higher-dimensional features. For instance, if the proposed feature was 3-dimensional, one could project onto different orthogonal lines and planes and measure the mutual information between the projections. However, in higher dimensions, both optimizing over all possible combinations of orthogonal lines and hyperplanes and computing the mutual information between higher-dimensional features may be challenging. Thus, in practice, we average the separability scores for adjacent pairs of PCA dimensions; this is not as principled but is more computationally tractable.
>
> > Hypothesis 2: while this is a stylistic point, I think the use of a possessive in "Our Superposition Hypothesis" is unnecessary and un-scientific. It could have any other objective name, say, the "Multi-Dimensional Superposition Hypothesis". See, e.g., Knuth et al., page 2 item 6.
>
> We agree and have updated the name to your suggestion, thank you!
>
> > p. 5, lines 260--270: doesn't this claim depend heavily on (1) the two-dimensional feature being active in the LLM's representation space and (2) the SAE is large enough to capture this feature? I'm not even sure if this claim is necessary, as opposed to saying "some of SAE clusters represent irreducible features".
>
> Thank you for pointing this out, we agree that it does rely on these things and have updated the text to make this more clear. We still think it is useful to provide intuition for why we might expect clusters of SAE features to represent multidimensional features. Does the updated text make this more clear?
>
> > Figure 3: what are some of the other features that are shown here to be more reducible than the features for Fig. 1?
>
> We have added a selection of these in Figure 16, and an accompanying discussion in Appendix G. These clusters seem to mostly represent letters or common words. We hypothesize these may perhaps be related to the “belief states” discovered by Shai et. al. [1], although we acknowledge more work is needed to understand them.
>
> > Are there any other features that we may intuitively view as circular but not found by the SAE? Say, are color words (red-orange-yellow-green-blue-purple) represented in a circular fashion?
>
> This is an excellent question; we have not investigated this, but we suspect there are some, as SAEs are known to have trouble representing continuous quantities. We have added some discussion of this to our conclusion.

---

> ### Author Response · Authors · 2024-11-19
> **Response Part 3**
>
> **Responses to questions, part 2**:
>
> > p. 8, "In practice, ...": this is somewhat confusing/concerning to me. If there are alternative pathways that may affect both the circular feature and the task output, then wouldn't they "wash away" the causal implication of the feature? In other words, in what sense exactly is this a "causal" implication rather than correlational?
>
> Could you clarify what you mean here? Certainly average ablating the other dimensions is a weaker form of subspace patching then just patching the circle and leaving the other dimensions unchanged, but it still seems that is is fundamentally a causal intervention. The idea is to show that limiting the days of the week representation to the circle maintains almost all of the performance for the task and results in an interpretable representation we can intervene on.
>
> > Broad question: are there examples of multi-dimensional features that aren't circular?
>
> Yes, see for example this recent work from Shai et. al. finding fractal belief states [1] (although in a toy transformer trained on a Hidden Markov Model) and additional recent work finding a spherical globe in an LLM [2]. There is also recent work on “Onion-like” nested features (although in a toy RNN trained on regression). All of these representations are irreducible multi-dimensional features by our definitions. We have added a discussion of [1] to the related work, and will add a discussion of the “Onion-like” features in the non-anonymous version of the paper (they cite our work, so citing it here would break anonymity).
>
> [1] https://www.lesswrong.com/posts/gTZ2SxesbHckJ3CkF/transformers-represent-belief-state-geometry-in-their
>
> [2]
> https://www.lesswrong.com/posts/guNzr32FC6DYCaSgJ/there-is-a-globe-in-your-llm
>
> If our comments resolve your concerns, we would appreciate if you would consider raising your score!

---

> > ### Comment · Reviewer_x6ai · 2024-11-24
> > **Helpful and thorough clarifications**
> >
> > Thank you for your helpful and thorough response. These responses and your proposed edits strengthen the overall paper. I have updated my rating to reflect the edits.
> >
> > What I meant by the "wash away" comment was how exactly (or at least intuitively) averaging out contributes to isolating the causal effect. It's not obvious to me what the averaging has to do with the causal effect. If this could be further clarified in the final edits, that'd be great.
> >
> > It's great to see having the limitations section & clarifying the precise claims of the paper, as it helps with the overall merits of the paper. A gentle suggestion for the final edits is to expand upon some of these points in a bit more detail, as they are interesting/intriguing (not further experiments or anything, but just some more thoughts). For example, if it's "unclear if this theory provides the best description for the representations models use," which I understand, it'd help the reader if you can imagine what kinds of further evidence would strengthen or weaken the claim (say, ones you couldn't test in the paper).

---

> > > ### Author Response · Authors · 2024-11-26
> > >
> > > > What I meant by the "wash away" comment was how exactly (or at least intuitively) averaging out contributes to isolating the causal effect. It's not obvious to me what the averaging has to do with the causal effect. If this could be further clarified in the final edits, that'd be great.
> > >
> > > The intuition behind averaging is to avoid “backup circuits” in the model from interfering with our intervention (see [1] for discussion of this backup behavior). For example, if there is some other subspace that encodes days of the week in a different way, if we intervene only on the circle subspace this additional subspace will “contradict” with the days of the week subspace, and if the downstream algorithm “averages” results from a few different circuits, then this may wash out the effect of our intervention or even put the model off distribution. We have clarified our discussion of this point in section 5.1, please let us know if it remains unclear.
> > >
> > > > It's great to see having the limitations section & clarifying the precise claims of the paper, as it helps with the overall merits of the paper. A gentle suggestion for the final edits is to expand upon some of these points in a bit more detail, as they are interesting/intriguing (not further experiments or anything, but just some more thoughts). For example, if it's "unclear if this theory provides the best description for the representations models use," which I understand, it'd help the reader if you can imagine what kinds of further evidence would strengthen or weaken the claim (say, ones you couldn't test in the paper).
> > >
> > > This is a great suggestion, thank you! We have added a short sentence at the end of the conclusion proposing future work we believe might shed light on some of the questions we leave off with.
> > >
> > > Thank you again, and let us know if you have any other questions!
> > >
> > > [1] Interpretability in the Wild: a Circuit for Indirect Object Identification in GPT-2 small, Wang et. al.

---

### Author Response · Authors · 2024-11-19
**Title Change**

Thank you all for the excellent and helpful comments! We have now responded to each reviewer with specific comments and have updated the PDF with the suggested changes, please let us know if anything is still unclear.

**Clarifying note**: We have changed the title of our paper from "Not All Language Model Features Are Linear" to "Not All Language Model Features Are One-Dimensionally Linear," as suggested by Reviewer 9dQp.

---

### Author Response · Authors · 2024-11-24

Friendly reminder to reviewers that if you have any questions, our window for responding to them will close soon! We hope we’ve adequately addressed your questions and concerns about the paper, thank you again for all of your excellent comments!

---

### Meta-Review · Area_Chair_3zeS · 2024-12-20

**Metareview:**

This paper shows that there are some concepts in large language models that are naturally represented as elements of low dimensional subspaces with dimension greater than 1. This fits into a broader line of work formalizing and extending the linear representation hypothesis in language models. Although the paper in its original state seems to have been somewhat confusing in how it presented its results, this seems adequately addressed following discussion with the reviewers. I would nevertheless urge the authors to further contextualize this paper in terms of related work (e.g., https://arxiv.org/abs/2406.01506 and https://arxiv.org/abs/2408.10920 off the top of my head, but there's a significant literature in this space).

**Additional Comments On Reviewer Discussion:**

see above

---

### Decision · Program_Chairs · 2025-01-22

Accept (Poster)